# Decoupling Planning from Control:
# Stable Hierarchical RL with a Learned Metric Space

**Sho Mitsuhashi[1]**                                                          *mitsuhashi.sho.7c@kyoto-u.ac.jp*

**Shin Ishii[1,2,3]**                                                               *ishii@i.kyoto-u.ac.jp*

[1] *Graduate School of Informatics, Kyoto University*
[2] *Department of AI-Brain Integration, Advanced Telecommunications Research Institute International*
[3] *International Research Center for Neurointelligence, The University of Tokyo*

**Reviewed on OpenReview:** *https://openreview.net/forum?id=Kmtlv8XOBN*

## Abstract

Hierarchical Reinforcement Learning (HRL) offers a promising framework for solving complex, long-horizon tasks by decomposing them into manageable subproblems. However, conventional HRL methods suffer from a critical non-stationarity problem: the high-level planner's learning process is destabilized because the low-level policy is concurrently learning and constantly changing. This issue is particularly severe in resource-constrained systems, such as edge-cloud robotics, where the low-level controller must be a computationally simple, low-capacity model. To address this challenge, we propose a novel HRL framework that mitigates the non-stationarity issue by decoupling high-level planning from low-level control. The core of our approach is to reframe the planner's task: instead of learning the planner via RL on non-stationary transitions, it learns to navigate a learned "map" of the environment. This map is represented by a critic network trained to function as a metric space, where distances reflect approximate travel costs. Planning is then simplified to finding optimal subgoals that lie along the shortest path (geodesic) between the current state and the final goal. To further encourage geometric consistency in the learned map, we introduce a trajectory regularization loss based on the agent's experienced trajectories. Experiments demonstrate that our decoupled framework is highly robust. In scenarios with resource-constrained low-level policies, our method learns to solve complex tasks effectively where standard approaches fail. This result highlights our framework's suitability for real-world systems where low-level controllers have inherently limited computational capacity. Our code is available at `https://github.com/sho-mitsuhashi/decoupled-hrl`.

## 1 Introduction

Autonomous agents that can solve complex, long-horizon tasks are a central goal in artificial intelligence, spanning domains from robotics to navigation (Kober et al., 2013; Zhu et al., 2017). While reinforcement learning (RL) provides a powerful paradigm for training such agents, standard methods often suffer from poor sample efficiency, long-horizon credit assignment, and intractable exploration under sparse rewards (Yu, 2018; Vezhnevets et al., 2017; Sutton et al., 1998).

**Hierarchical Reinforcement Learning (HRL)** offers a principled way to mitigate these challenges by decomposing difficult problems into simpler subproblems via state and temporal abstraction (Sutton et al., 1999; Dayan & Hinton, 1992). A high-level policy (the *planner*) proposes intermediate subgoals, and a low-level policy (the *worker*) executes goal-conditioned control to realize each subgoal over short horizons. This hierarchical structure yields several benefits: improved credit assignment by feedback on subgoal achievement (Vezhnevets et al., 2017; Florensa et al., 2017); structured, goal-directed exploration that narrows

the search space (Kulkarni et al., 2016; Machado et al., 2017); and increased interpretability of agent behavior (Andreas et al., 2017; Krishnan et al., 2017). Collectively, these properties make HRL a promising direction toward scalable, data-efficient learning on long-horizon tasks.

Intermediate subgoals are particularly useful because they separate global routing from local control. A direct goal-conditioned policy must handle both simultaneously, which can be especially demanding in the long-horizon and resource-constrained settings considered in this paper. By contrast, subgoals reduce this burden by shortening the worker's control horizon while allowing the planner to capture global path structure.

Despite its appeal, HRL faces practical challenges that have hindered its widespread adoption. The primary obstacle is **the problem of non-stationarity** (Gürtler et al., 2021; Hutsebaut-Buysse et al., 2022), where the high-level policy learns under effectively time-varying dynamics because the low-level controller is continuously updated. As a result, even for the same subgoal, the controller's behavior and induced state transitions can differ across training, destabilizing high-level learning.

The root of this problem lies in the "end-to-end concurrent learning" paradigm emphasized by recent HRL approaches. While early HRL research avoided non-stationarity by using pre-trained and fixed low-level skills, this lacked flexibility and generality (Stolle & Precup, 2002; Menache et al., 2002). Consequently, methods like HAC (Hierarchical Actor-Critic) and HIRO (Hierarchical Reinforcement learning with Off-policy correction) aimed to learn all policies within the hierarchy in parallel and integrally (Levy et al., 2019; Nachum et al., 2018). However, this integrated learning becomes the cause of non-stationarity. The state transition probability from the planner's view, $P(s'|s, \text{subgoal})$, is determined by the low-level policy's behavior. As the low-level policy changes through learning, this transition probability also changes, rendering the high-level's MDP non-stationary. Thus, these methods face a fundamental dilemma: the very feature that makes them powerful also creates their greatest weakness.

This non-stationarity problem is not merely an academic issue; it becomes more pronounced when attempting to apply HRL to real-world applications, particularly in distributed systems like **Edge-Cloud computing** (Tahir & Parasuraman, 2025; Reusch et al., 2023).

The Edge-Cloud architecture is becoming a standard configuration in robotics and IoT (Pereira et al., 2020). In this model, a resource-constrained robot or sensor (the edge device) handles physical interaction and low-level motor control, while a computationally powerful cloud server manages data analysis and high-level strategic decision-making (planning) (Tahir & Parasuraman, 2025).

Figure 1: A conceptual diagram of our decoupled HRL architecture, designed to mitigate planner instability caused by worker-induced transitions. In the Training Phase, a learned "map" of the environment is constructed by learning a latent metric space from the agent's experience trajectories; in this space, distances represent approximate travel costs. The High-Level Planner is trained to find subgoals using this learned geometry, rather than learning from worker-induced transitions. During the Execution Phase (top), the trained planner proposes a Subgoal between the current State and the final Goal. The low-level Worker then executes this subgoal.

This architecture naturally corresponds to the HRL hierarchy: a worker (low-level policy) running on the edge device and a planner (high-level policy) on the cloud. However, a critical constraint exists: edge devices are under strict limitations in terms of CPU performance, memory, and power consumption (Li et al., 2018). Therefore, the worker's policy network must be an extremely lightweight and computationally inexpensive model.

This constraint exacerbates the non-stationarity problem. A simple, low-capacity worker is highly unstable, forcing the cloud-based planner to learn from extremely noisy information. This **tight coupling**—the direct dependency of the planner on the worker's volatile performance—is a key limitation of conventional HRL. As our analysis in Appendix B reveals, this conventional paradigm performs poorly in the resource-constrained Edge-Cloud scenario. This failure clarifies the essential requirement for a new solution: the planner should be less sensitive to the worker's performance fluctuations. Such a real-world demand motivates our proposed decoupling methodology.

To address the non-stationarity problem in HRL, this paper proposes a paradigm shift: **Decoupling high-level planning from low-level control**. The core of this approach is to switch the planner's dependency from the changing worker policy to a critic-defined geometric map of the environment. This critic is learned to define a (non-Euclidean) metric-structured space—an abstract map where the "distance" between two points approximates travel cost (Pitis et al., 2020).

The technical foundation for this idea is the **Metric Residual Network (MRN)** (Liu et al., 2023). This framework is built upon an important geometric property of the optimal action-value function $Q^*(s, a, g)$ in Goal-Conditioned Reinforcement Learning (GCRL). Specifically, it theoretically proves that the negative optimal value function, $-Q^*$, exhibits key properties of a distance metric, satisfying the triangle inequality in a specific sense.

The **triangle inequality**, i.e., the property that for any three points $x, y, z \in \mathbb{R}^D$, $d(x, z) \leq d(x, y) + d(y, z)$, is a fundamental property of distance and the cornerstone of our subgoal planning. An intermediate point $y$ that satisfies equality can be viewed as a desirable subgoal, since it lies along the shortest path between $x$ and $z$. We utilize this property for planning.

Figure 1 provides an overview of the proposed framework. Rather than learning directly from unstable worker-induced transitions, the planner is trained to navigate a stable learned map induced by the critic. This is particularly useful when the final goal is too distant to serve as an effective target for a low-capacity worker: the planner instead proposes intermediate subgoals along geodesics (shortest paths) in the learned metric space. Because this metric captures the environment's intrinsic travel costs, it provides a stable learning objective that is substantially less sensitive to worker volatility.

This paper introduces a novel HRL framework that mitigates non-stationarity by applying insights from metric learning. Our **contributions** are summarized in the following three points:

1. **A decoupled HRL framework to mitigate non-stationarity:** We propose a novel framework where the high-level planner learns from a stable, learned "map" of the environment, rather than being trained *via RL* on the unstable worker-induced transition dynamics. This map is a critic network trained as a metric space where distances approximate travel costs. By avoiding direct planner updates on worker-induced transitions, the framework reduces non-stationarity in high-level learning and stabilizes learning.

2. **Trajectory regularization for geometric consistency:** We introduce a regularization loss that encourages geometric consistency along the agent's experienced trajectories. This data-driven term is intended to refine the learned metric space used by the planner.

3. **Robustness under resource-constrained workers:** We experimentally show that our framework enables agents to solve complex tasks even with a low-capacity worker policy—a scenario where standard non-hierarchical agents fail and, as demonstrated in Appendix B, conventional hierarchical agents become substantially less effective. This highlights its suitability for real-world systems with hardware limitations, such as Edge-Cloud robotics.

## 2   Related Work

**Geometric structure in goal-conditioned RL.**   A growing line of work studies goal-conditioned value functions through the lens of geometry. In sparse-reward goal-reaching problems, the negative optimal goal-conditioned value can be interpreted as a distance-like quantity, motivating architectures and objectives

that explicitly encode quasimetric structure. Metric Residual Networks (MRN) parameterize the critic so that its negative output behaves as a quasimetric, improving sample efficiency in goal-conditioned RL (Liu et al., 2023). Quasimetric RL (QRL) further develops this view by introducing learning objectives tailored to quasimetric value functions and providing corresponding theoretical analysis (Wang et al., 2023). More broadly, triangle-inequality-respecting neural distance models have also been studied as an inductive bias for learning structured distances (Pitis et al., 2020). Our work builds on this geometric perspective, but differs in purpose: rather than using the critic only as a better value estimator, we use the learned geometry as the substrate for *hierarchical planning*.

**Subgoal-based hierarchical RL and non-stationarity.** Subgoal-based HRL methods learn a high-level policy that proposes intermediate subgoals and a low-level policy that executes them. Representative methods such as HAC and HIRO showed that such hierarchies can improve long-horizon learning while learning multiple levels jointly, but also revealed a central challenge: concurrent low-level learning makes the effective dynamics seen by the high-level policy non-stationary (Levy et al., 2019; Nachum et al., 2018). Subsequent work introduced additional stabilization mechanisms. For example, HiTS reduces non-stationarity by introducing timed subgoals that modify the induced high-level process and its resulting semi-Markov dynamics (Gürtler et al., 2021). PIPER instead stabilizes high-level learning through reward relabeling via a learned reward model and feasibility-aware regularization, encouraging subgoals that remain achievable under the current low-level policy (Singh et al., 2024). In contrast, our method addresses this dependence more directly by optimizing the planner against a critic-defined geometric objective, rather than learning it as an RL policy over worker-induced high-level transitions.

**Value-based hierarchy and planning over learned structure.** Our method is also related to recent work that derives hierarchy or planning mechanisms from learned goal-conditioned value structure. HIQL learns hierarchical goal-conditioned behavior from offline data by extracting both high- and low-level policies from a single value function, using latent states as high-level actions (Park et al., 2023). This is closely related in spirit to our use of value structure for hierarchy. However, HIQL is primarily motivated by the difficulty of learning policies from noisy value estimates for distant goals in the *offline* setting, whereas our focus is *online* hierarchical RL and stabilizing learning by decoupling planning from worker-dependent transitions. Our approach is also related to planning-based methods such as SoRB and HIGL, which use explicit planning to identify intermediate waypoints or landmarks to guide long-horizon subgoal generation (Eysenbach et al., 2019; Kim et al., 2021). In contrast, rather than relying on an explicit search procedure, we learn a parametric planner that proposes subgoals directly in the critic's learned geometric space.

**Positioning of this work.** Our method lies at the intersection of geometric value-function learning, subgoal-based hierarchical RL, and planning over learned goal-conditioned structure. Relative to MRN and QRL, our contribution is not a new quasimetric critic or RL objective, but a hierarchical use of goal-conditioned geometry as a planning map. Relative to HIRO, HAC, HiTS, and PIPER, our key difference is that the planner is not trained through worker-induced high-level transitions, reducing a central source of non-stationarity in online HRL. Relative to HIQL, we also exploit learned goal-conditioned value structure for hierarchy, but focus on the online setting rather than offline goal-conditioned RL. Relative to SoRB and HIGL, we use learned goal-conditioned structure for long-horizon control, but with a learned planner rather than explicit search.

## 3 Preliminaries

### 3.1 Goal-Conditioned Reinforcement Learning

Goal-Conditioned Reinforcement Learning (GCRL) extends the standard RL framework by training an agent to achieve a variety of goals (Schaul et al., 2015; Nair et al., 2018). The agent's policy $\pi(a_t \mid s_t, g)$ is conditioned not only on the current state $s_t \in \mathcal{S}$ but also on a goal $g \in \mathcal{G}$. At the beginning of each episode, an initial state $s_0 \sim \rho_0$ and a target goal $g \sim \rho_G$ are sampled, and the agent interacts with the environment to maximize the expected discounted return.

Formally, GCRL is modeled as a goal-conditioned Markov decision process:

$$\mathcal{M}_{\mathrm{gc}} = (\mathcal{S}, \mathcal{A}, \mathcal{G}, T, R, \gamma, \rho_0, \rho_G),$$

where $\mathcal{S}$, $\mathcal{A}$, and $\mathcal{G}$ denote the state, action, and goal spaces, $T(\cdot \mid s, a)$ denotes the transition distribution over next states, $R : \mathcal{S} \times \mathcal{A} \times \mathcal{G} \to \mathbb{R}$ is a scalar-valued reward function, $\gamma \in (0, 1)$ is the discount factor, and $\rho_0, \rho_G$ are the initial-state and goal distributions, respectively.

A common reward structure in this setting is a sparse reward. In the experiments of this paper, unless otherwise noted, we employ the *standard sparse reward* commonly used in GCRL:

$$r_{t,g} = R(s_t, a_t, g) = \begin{cases} 0 & \text{if } M(s_t, a_t) = g, \\ -1 & \text{otherwise.} \end{cases}$$

This reward provides $0$ upon goal achievement and $-1$ otherwise. The deterministic onto mapping $M : \mathcal{S} \times \mathcal{A} \to \mathcal{G}$ is introduced to handle the general case where $\mathcal{G} \neq \mathcal{S}$ (e.g., success may require a terminal action).

The universal action-value function (UVFA) is then defined as (Schaul et al., 2015):

$$Q^\pi(s, a, g) = \mathbb{E}_\pi \left[ \sum_{t=0}^{\infty} \gamma^t r_{t,g} \;\middle|\; s_0 = s, \ a_0 = a, \ g \right],$$

which is the expected discounted return to reach goal state $g$ at state $s$ by choosing action $a$.

## 3.2 Hierarchical Reinforcement Learning

Hierarchical Reinforcement Learning (HRL) is a method designed to solve complex, long-horizon problems by decomposing them into smaller, more manageable sub-problems. Instead of learning a single, monolithic policy to navigate from a starting state to a distant final goal, HRL employs a multiple-level hierarchy to make learning more efficient and tractable. This hierarchy consists of one or more high-level policies and a low-level policy.

- **High-Level Policy** $\pi_{hl}(subgoal|s_t, g)$: Often considered the "planner," the high-level policy is responsible for strategic, long-range decision-making. Its role is to observe the current state $s_t$ and the final goal $g$, and then to set an appropriate intermediate subgoal $subgoal \in G$. The objective of this policy is to choose a subgoal that will guide the agent toward the final goal.

- **Low-Level Policy** $\pi_{ll}(a_t|s_t, subgoal)$: The low-level policy, or "worker", is responsible for executing these subgoals. Its task is to receive a specific subgoal from the high-level policy and generate a primitive action to reach that subgoal from the current state.

## 3.3 The Triangle Inequality in Goal-Conditioned RL

A key geometric insight from Liu et al. (2023), which underpins our approach, is that in goal-conditioned RL (GCRL) with sparse, non-positive rewards, **the negative of the optimal universal action-value function, $-Q^\star$, satisfies the triangle inequality**, a fundamental property of distance metrics.

This principle is most clearly demonstrated in a setting where the goal space is equivalent to the state-action space, i.e., $\mathcal{G} \equiv \mathcal{S} \times \mathcal{A}$. Let the state-action space be $\mathcal{X} = \mathcal{S} \times \mathcal{A}$, with elements $x = (s, a)$. The value function $Q^\star(s, a, g)$ can then be written as $Q^\star(x, x_g)$, where $x_g$ is the goal element specified by state-action pair. For any three points $x_1, x_2, x_3 \in \mathcal{X}$, the optimal value function is constrained as follows:

$$Q^\star(x_1, x_2) + Q^\star(x_2, x_3) \leq Q^\star(x_1, x_3).$$

Note that multiplying this inequality by $-1$ transforms this into the standard form of the triangle inequality. Intuitively, the total reward to get from $x_1$ to $x_3$ is greater than the total reward of going via an intermediate

point $x_2$. This geometric structure is also preserved in the more general setting where goal is not specified by $\mathcal{S} \times \mathcal{A}$.

However, $-Q^\star$ is not a true metric. It can be asymmetric, meaning the cost to travel from $x_1$ to $x_2$ is not necessarily the same as from $x_2$ to $x_1$. This structure is more accurately described as a **quasipseudometric**[1], which satisfies the triangle inequality but does not require symmetry.

### 3.4 Metric Residual Network

Motivated by the structure above, the *Metric Residual Network* (MRN) parameterizes the critic so that its *negative* output is, by construction, a quasipseudometric in a learned latent space (Liu et al., 2023). Concretely, MRN has two stages:

**(i) Projection to a shared latent space.** Two encoders map inputs to latent vectors

$$z_1 \;=\; e_1(s, a) \in \mathbb{R}^d, \qquad z_2 \;=\; e_2(s, g) \in \mathbb{R}^d,$$

where $e_1$ and $e_2$ are multi-layer perceptrons (MLPs), and $d$ is the dimension of the latent space. The motivation for this decomposition is to separate the inputs into two contexts: $e_1(s, a)$ encodes the current **state–action context**, while $e_2(s, g)$ encodes the **goal-conditioned context**.

**(ii) Symmetric + asymmetric distance.** To enforce the quasipseudometric structure in the latent space, MRN further decomposes the latent distance into a symmetric metric term and an asymmetric residual. Define

$$d_{\mathrm{sym}}(z_1, z_2) = \|\phi(z_1) - \phi(z_2)\|_2\,, \qquad d_{\mathrm{asym}}(z_1, z_2) = \max_{i \in [K]} \big[h_i(z_1) - h_i(z_2)\big]_+,$$

where $\phi$ and $h_i$ are learnable MLPs, and $[\cdot]_+$ denotes the ReLU function. The critic is then

$$Q_\theta(s, a, g) \;=\; -\Big(d_{\mathrm{sym}}(z_1, z_2) \;+\; d_{\mathrm{asym}}(z_1, z_2)\Big).$$

By construction, $d = d_{\mathrm{sym}} + d_{\mathrm{asym}}$ is nonnegative, satisfies $d(x, x) = 0$, and obeys the triangle inequality; thus $-Q_\theta$ **is a quasipseudometric on the latent space**. This is an *architectural property* of the MRN parameterization, not a claim that the critic has already converged to $Q^\star$: for any parameters $\theta$, the negative MRN output defines a quasipseudometric in the learned latent space, while learning determines how well this geometry approximates the true optimal travel cost. Moreover, MRN *universally approximates* any continuous quasipseudometric on a compact domain, ensuring sufficient expressivity to represent $-Q^\star$.

This decomposed architecture provides an effective inductive bias for approximating the universal action-value function. In particular, the symmetric part encourages sample-efficient approximation through the $\ell_2$ term, while the asymmetric part captures directional difficulty that the $\ell_2$ term alone cannot represent (Liu et al., 2023).

## 4 Metric-based Hierarchical Reinforcement Learning

We now present our two contributions: (i) a decoupled hierarchical framework in which the high level policy plans purely in the critic's distance space, and (ii) a trajectory regularizer that explicitly enforces near-equality in the triangle inequality along observed agent trajectories.

### 4.1 Decoupled Planning on a Learned Metric Map

Our core innovation is to **decouple** the high-level planner from the unstable, concurrently trained low-level worker. While conventional HRL learns the planner via RL on worker-induced transitions, our planner

---

[1]A quasipseudometric also requires identity $d(x, x) = 0$. This property is satisfied by conceptually augmenting the state space.

does not optimize against those non-stationary transitions; it performs *pure planning* in the critic's learned distance space. The critic serves as a stable metric "map" where distances approximate optimal travel cost.

The fundamental principle guiding our planner is simple geometry: an optimal subgoal must lie on the geodesic (i.e., the shortest path) between the current state and the final goal. Crucially, this is not a simple Euclidean map where the shortest path is a straight line. Instead, the **critic learns a non-Euclidean metric space** that respects the environment's true topology, such as walls and other obstacles. This principle is motivated by the triangle-inequality structure of the optimal goal-conditioned value function established by Liu et al. (2023). In our method, this geometry is instantiated by the MRN-based critic, whose negative output is designed to satisfy a quasipseudometric structure. Accordingly, for the optimal value function, the relation for any state $s$, subgoal $subgoal$, and goal $g$ can be written as follows, where action arguments are omitted for simplicity:

$$-Q^\star(s, g) \le -Q^\star(s, subgoal) + -Q^\star(subgoal, g)$$

.

As illustrated in Figure 2, this means the cost of the direct path is always less than or equal to the cost of any two-step path via a subgoal. An optimal subgoal is one that lies on the shortest path, turning this inequality into a near-equality. Therefore, the planner's objective is to find a subgoal that maximizes the value of the two-step path, bringing it as close as possible to the value of the direct path.

However, directly searching for such a subgoal in the raw state or goal space is challenging, as these spaces may differ in dimension and structure (e.g., state includes velocity, while goal is only position). To circumvent this, our planner operates entirely within the unified latent space learned by the MRN critic.

As we discussed in Sec 3.4, the critic encodes the current state-action context into a latent vector $z_1 = e_1(s, a)$ and the goal context into $z_2 = e_2(s, g)$. Our high-level policy, $\pi_{\text{hl}}$, then takes these two latent points and proposes an intermediate latent subgoal, $z_{sub} = \pi_{\text{hl}}(z_1, z_2)$.

The high-level policy is trained to maximize the following objective function, which directly formalizes the geometric principle described above:

$$J_{\text{hl}}(\theta; s, a, g) = Q_\theta(z_1, z_{sub}) + Q_\theta(z_{sub}, z_2), \quad (1)$$

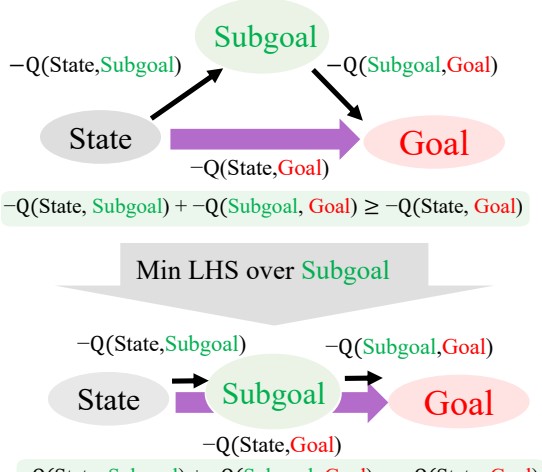

Figure 2: A geometric approach to optimal subgoal selection. The triangle inequality, applied to the value function (top), shows that the cost of any two-step path via a subgoal is lower-bounded by the cost of the direct path. The planner's objective is to find a subgoal that minimizes the left-hand side of this inequality. An optimal subgoal, which lies on the geodesic (shortest path), makes this inequality tight (bottom). Action arguments are omitted for simplicity.

Here, $Q_\theta(z_a, z_b)$ is a shorthand for the value computed by the MRN's internal distance function between the latent contexts $z_a$ and $z_b$. Since this structure respects the triangle inequality, $J_{\text{hl}}$ is always upper-bounded by $Q_\theta(z_1, z_2)$, the value of the direct path.

Importantly, the high-level policy is trained by maximizing Equation 1, rather than via RL on worker-induced high-level transitions as in conventional HRL. Although the critic $Q_\theta$ is learned jointly and changes during training, the resulting non-stationarity is qualitatively different. In conventional HRL, updates to the low-level worker directly change the effective high-level transition process faced by the planner, so the planner is trained on an evolving MDP. In our method, the planner instead optimizes a geometric objective on the current critic for the same underlying goal-conditioned control problem. Low-level updates therefore affect the planner only indirectly through the replay data used to improve the critic, rather than by directly changing its transition dynamics. Thus, our method avoids direct worker-induced transition non-stationarity

at the planner level. A control experiment in Appendix D further suggests that the gains of our method cannot be explained simply by freezing the low-level system.

At execution time, the planned latent subgoal $z_{sub}$ is mapped back to a concrete goal $\tilde{g}_{sub} = D_\psi(z_{sub})$ by a lightweight decoder, which is then passed to the low-level policy $\pi_{ll}(\cdot \mid s, \tilde{g}_{sub})$.

Finally, this formulation naturally extends to multi-step planning by recursively applying the **same** policy $\pi_{hl}$. See Appendix A for more details.

## 4.2 Data-Driven Trajectory Regularization

In addition to the standard TD learning objective, we introduce a trajectory regularization loss to train the critic $Q_\theta$. The goal of this regularizer is to encourage the geometric consistency of the value function along observed agent trajectories. The core intuition is that trajectories sampled from the replay buffer, especially those from later stages of training, serve as reasonable approximations of the true geodesic paths within the environment. By encouraging the critic's learned metric to conform to these experienced paths, this regularizer helps provide a more useful geometric basis for high-level planning.

To formalize this, we sample trajectories of states and actions $(s_i, a_i)_{i=0}^T$ from the replay buffer. Following the principle of Hindsight Experience Replay (HER) (Andrychowicz et al., 2017), we treat the state $s_t$ that the agent actually reached at time $t$ as the "achieved goal" $g_t$ for that step. For any triplet of time steps $(i < j < k)$ along this trajectory, we use the achieved goals $g_j$ and $g_k$ to construct our regularization loss. The loss penalizes deviations from the triangle *equality*, where the expectation $\mathbb{E}$ is taken over trajectories sampled from the replay buffer:

$$\mathcal{L}_\triangle(\theta) = \mathbb{E}\left[\frac{(Q_\theta(s_i, a_i, g_j) + Q_\theta(s_j, a_j, g_k) - Q_\theta(s_i, a_i, g_k))^2}{(\#\text{Timesteps from } s_i \text{ to } s_k)^2}\right].$$

The total critic loss is a weighted sum of the standard TD loss and our regularization loss:

$$\mathcal{L}_{\text{critic}}(\theta) = \mathcal{L}_{\text{TD}}(\theta) + \lambda\mathcal{L}_\triangle(\theta). \tag{2}$$

The denominator serves as a normalization term; since the squared error in the numerator is expected to scale with the square of the elapsed time steps, this term ensures the loss's magnitude remains consistent regardless of the temporal separation between states. This stabilizes the learning process and makes the framework less sensitive to the choice of the hyperparameter $\lambda$. In all of our experiments, we set $\lambda = 1$. Ultimately, this regularization encourages geometric consistency of the value function along experienced trajectories, improving the learned geometry used by the high-level planner.

## 4.3 Training Procedure

Our framework consists of four main neural network components: the MRN critic $Q_\theta$, the high-level policy $\pi_{hl}$, the low-level policy $\pi_{ll}$, and a goal decoder $D_\psi$. These components are trained jointly in an off-policy manner using an experience replay buffer populated with agent trajectories. Each training step involves updating the four components based on a minibatch of transitions sampled from the buffer, as detailed below.

**Low-Level Policy Training.** The low-level policy $\pi_{ll}$ (with parameters $\phi_{ll}$) is trained as a standard goal-conditioned agent. Its objective is to learn a policy that can reach any given goal $g \in \mathcal{G}$. For a given state $s$, it learns to produce an action $a = \pi_{ll}(s, g)$ that maximizes the critic's Q-value. The policy is updated by minimizing the DDPG actor loss (Lillicrap et al., 2016):

$$\mathcal{L}_{ll} = -\mathbb{E}_{s,g\sim\mathcal{D}}\left[Q_\theta(s, \pi_{ll}(s, g), g)\right]. \tag{3}$$

Crucially, the goals $g$ used in this update are sampled from the replay buffer and relabeled using Hindsight Experience Replay (Andrychowicz et al., 2017), not the subgoals generated by the high-level policy. This design choice is critical for stable learning, as it prevents the low-level policy from being misguided by potentially inaccurate subgoals, especially in the early stages of training.

**High-Level Policy Training.** The high-level policy $\pi_{\mathrm{hl}}$ is responsible for planning in the latent space. It is trained to maximize the objective $J_{\mathrm{hl}}$ in Equation 1 by applying gradient ascent. During this update, the parameters of the critic, low-level policy, and decoder are temporarily frozen, so that only the high-level policy is updated against the current critic-defined geometric structure. This freeze is temporary and local to the current high-level update. A key feature of our approach is that a single, shared $\pi_{\mathrm{hl}}$ can be used for multi-step planning. See Appendix A for multi-step cases.

**Goal Decoder Training.** To enable the planner's latent subgoals to be decoded into concrete subgoals during execution, the goal decoder $D_\psi$ learns to map the latent goal representations back to the original goal space $\mathcal{G}$. It is trained to reconstruct the original goal $g$ from its latent representation $z_2 = e_2(s, g)$, which is produced by the MRN's goal encoder. This training process, which minimizes a reconstruction error, is structurally identical to training the decoder of an Autoencoder. During this update, the parameters of the encoder $e_2$ are frozen. The decoder is optimized by minimizing the mean squared error:

$$\mathcal{L}_D(\psi) = \mathbb{E}_{s,g \sim \mathcal{D}} \left[ \|g - D_\psi(e_2(s, g))\|^2 \right]. \tag{4}$$

**Critic Training.** The critic $Q_\theta$ is updated as described in Section 4.2 by minimizing the combined loss $\mathcal{L}_{\mathrm{critic}}$ from Equation 2, which includes both the standard TD-error and our trajectory regularization loss.

The entire training process is summarized in Algorithm 1.

---

**Algorithm 1** Metric-Based Hierarchical RL Training

---

1: Initialize networks: $Q_\theta, \pi_{\mathrm{hl}}, \pi_{\mathrm{ll}}, D_\psi$.
2: Initialize target networks: $Q_{\theta'} \leftarrow Q_\theta, \pi_{\mathrm{ll}'} \leftarrow \pi_{\mathrm{ll}}$.
3: Initialize replay buffer $\mathcal{D}$.
4: **for** each epoch **do**
5:     Collect trajectories using current policies $(\pi_{\mathrm{hl}}, \pi_{\mathrm{ll}}, D_\psi)$ and store in $\mathcal{D}$.
6:     **for** each training step **do**
7:         Sample a minibatch of transitions from $\mathcal{D}$.
8:         Apply Hindsight Experience Replay to the minibatch.
9:         Update Critic $Q_\theta$ by minimizing $\mathcal{L}_{\mathrm{critic}}$ from Equation 2.
10:        Update $\pi_{\mathrm{ll}}$ by minimizing $\mathcal{L}_{\mathrm{ll}}$ from Equation 3.
           *— Update High-Level Policy —*
11:        Freeze $Q_\theta, \pi_{\mathrm{ll}}, D_\psi$.
12:        Update $\pi_{\mathrm{hl}}$ by maximizing $J_{\mathrm{hl}}$ from Equation 1.
13:        Unfreeze networks.
           *— Update Goal Decoder —*
14:        Freeze encoder $e_2$ within $Q_\theta$.
15:        Update $D_\psi$ by minimizing $\mathcal{L}_D$ from Equation 4.
16:        Unfreeze encoder $e_2$.
           *— Update Target Networks —*
17:        Update target networks with Polyak averaging:
18:        $\theta' \leftarrow \rho\theta' + (1 - \rho)\theta$
19:        $\phi'_{\mathrm{ll}} \leftarrow \rho\phi'_{\mathrm{ll}} + (1 - \rho)\phi_{\mathrm{ll}}$
20:     **end for**
21: **end for**

---

### 4.4 Execution-Phase Action Selection

During the execution phase, which includes both training data collection and final evaluation, the agent determines its action through a hierarchical planning process, as detailed in Algorithm 2. For any given time step $t$, the objective is to select an action $a_t$ based on the current state $s_t$ and the final goal $g_{\mathrm{final}}$.

---

**Algorithm 2** Execution-Phase Action Selection

---

1: **Input:** Current state $s_t$, final goal $g_{\text{final}}$
2: **Output:** Action $a_t$
   — *Encode final goal and current state context into latent space* —
3: $z_{\text{goal}} \leftarrow e_2(s_t, g_{\text{final}})$
4: $a_{\text{est}} \leftarrow \pi_{ll}(s_t, g_{\text{final}})$                       ▷ Pseudo-action to get state context
5: $z_1 \leftarrow e_1(s_t, a_{\text{est}})$
   — *Plan a single subgoal in latent space* —
6: $z_{\text{sub}} \leftarrow \pi_{hl}(z_1, z_{\text{goal}})$
   — *Decode subgoal and select action* —
7: $\tilde{g}_{\text{sub}} \leftarrow D_\psi(z_{\text{sub}})$
8: $a_t \leftarrow \pi_{ll}(s_t, \tilde{g}_{\text{sub}})$
9: **return** $a_t$

---

The core of this process is to leverage the high-level policy $\pi_{hl}$ to propose an intermediate subgoal that is situated on the optimal path toward the final goal. This planning occurs within the latent space learned by the critic. First, the final goal is encoded into its latent representation, $z_{\text{goal}} = e_2(s_t, g_{\text{final}})$.

To generate a subgoal that is reachable from the current state, the high-level planner also requires contextual information about the agent's current situation. This is provided by the state-action embedding $z_1 = e_1(s_t, a)$. However, since the action $a_t$ for the current step has not yet been determined, we estimate a plausible action $a_{\text{est}}$ by querying the low-level policy with the final goal: $a_{\text{est}} = \pi_{ll}(s_t, g_{\text{final}})$. This estimate allows us to compute the necessary state-action context embedding, $z_1 = e_1(s_t, a_{\text{est}})$.

With both the latent goal $z_{\text{goal}}$ and the latent state-action context $z_1$ available, the high-level policy generates a latent subgoal in a single step: $z_{\text{sub}} = \pi_{hl}(z_1, z_{\text{goal}})$. This latent subgoal is then decoded back into a concrete goal in the original goal space, $\tilde{g}_{\text{sub}} = D_\psi(z_{\text{sub}})$. Finally, this newly generated subgoal is passed to the low-level policy, which produces the action $a_t = \pi_{ll}(s_t, \tilde{g}_{\text{sub}})$ to be executed in the environment.

## 5 Experiments

We evaluate our method on a suite of standard Goal-Conditioned RL benchmark tasks to assess its performance across a diverse range of situations. Specifically, we use the robotics manipulation environments FetchPick, FetchPush, FetchSlide, and HandManipulateEggRotate from Plappert et al. (2018), alongside the PointMaze 2D-navigation environment from Fu et al. (2020). These tasks are selected to span *qualitatively different challenges* (topological complexity, dynamical complexity, and high dimensionality); full environment specifications are provided in Appendix C.

### 5.1 Compared Methods

To systematically evaluate the contributions of our proposed framework, we selected three methods for comparison in an ablation study. This approach allows us to isolate the impact of each component of our method.

- **Baseline: A standard goal-conditioned RL agent (using MRN) without hierarchy.**

  This method, based on the original MRN paper, serves as our primary control group. It is a non-hierarchical, goal-conditioned agent trained with DDPG and Hindsight Experience Replay. By comparing against this baseline, we can quantify the benefits of introducing a hierarchical structure.

- **Baseline + Hierarchy (Ours): Our proposed HRL framework using the metric-based subgoal selection but without the trajectory regularization.**

  This version isolates the core contribution of our work: the decoupling of high-level planning from low-level execution via a learned distance metric. Comparing this method to the baseline demonstrates the direct advantage of our hierarchical formulation, independent of other improvements.

- **Baseline + Hierarchy + Regularization (Ours): Our full proposed method.**

  This represents our complete framework, including the trajectory regularization term. The regularizer uses the agent's own experience trajectories to encourage geometric consistency in the learned metric space used for planning. Comparing this full method against the hierarchy-only version allows us to assess the empirical effect of the regularization term.

Additional comparisons and ablations are provided in the appendix, including a comparison with HIRO in Appendix B and an ablation examining the role of the MRN-based metric structure in Appendix F.

## 5.2 Evaluation Scenarios

We design our experiments to evaluate the setting most central to our claim: whether the proposed framework remains effective when the low-level controller is severely capacity-limited. We therefore begin with this regime as the primary evaluation setting and then consider a standard capable-worker setting as a complementary check.

- **Resource-Constrained Low-level Policy**: Our primary evaluation considers a worker with severely limited network capacity (2 layers, 8 hidden units), reflecting the practical Edge–Cloud setting that motivates this work. This scenario directly tests the core hypothesis of the paper: that the proposed decoupled hierarchy improves robustness and remains effective even when the low-level controller is simple and capacity-limited.

- **Capable Low-level Policy**: We also evaluate a standard setting in which the low-level policy is implemented with a high-capacity neural network (4 layers, 256 hidden units). This serves as a complementary check, verifying that the proposed hierarchy remains competitive and does not degrade performance when low-level capacity is sufficient.

## 5.3 Experimental Results

We begin with the resource-constrained worker setting, which is most central to the claim of the paper, and then report the capable-worker setting as a complementary check.

### 5.3.1 Performance with a Resource-Constrained Low-level Policy

This experiment directly tests the central hypothesis of the paper: that our decoupled framework provides robust performance even when the low-level control capacity is severely limited. This scenario simulates real-world constraints, such as those in an edge-cloud setting. The comparison against the non-hierarchical baseline is crucial for this analysis. It serves as a direct ablation study to quantify the specific benefits of our hierarchical planner, particularly its ability to guide a controller with limited capacity toward successful task completion.

The results, shown in Figure 3, reveal a clear performance difference between the methods. The non-hierarchical baseline agent, equipped with a low-capacity policy network, consistently underperformed the hierarchical variants across environments. These results suggest that the limited expressiveness of the policy can be a substantial bottleneck when the task is solved without hierarchical decomposition.

In contrast, both of our proposed hierarchical methods demonstrated effective learning, substantially outperforming the baseline. This is a key finding of our research: by decoupling high-level planning from low-level execution, our framework can generate effective intermediate subgoals that enable a simple, low-capacity policy to successfully solve complex tasks. The planner effectively compensates for the worker's limitations.

Furthermore, when comparing the two hierarchical variants, the method with trajectory regularization tended to perform better, with more noticeable gains in EggRotate and PointMaze. These results indicate that trajectory regularization may provide additional benefits in some resource-constrained settings, although its effect is not uniform across tasks. Overall, the results support the effectiveness of the proposed framework in the resource-constrained low-level control setting.

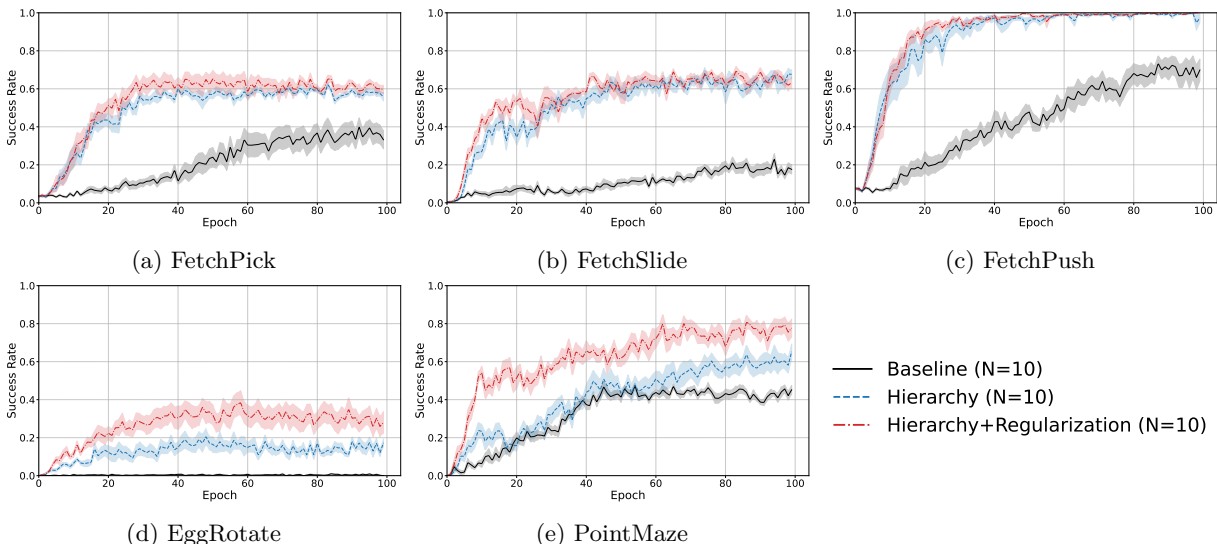

Figure 3: Mean success rate for the Resource-Constrained Low-level Policy setting. Shaded regions indicate the standard error over 10 random seeds.

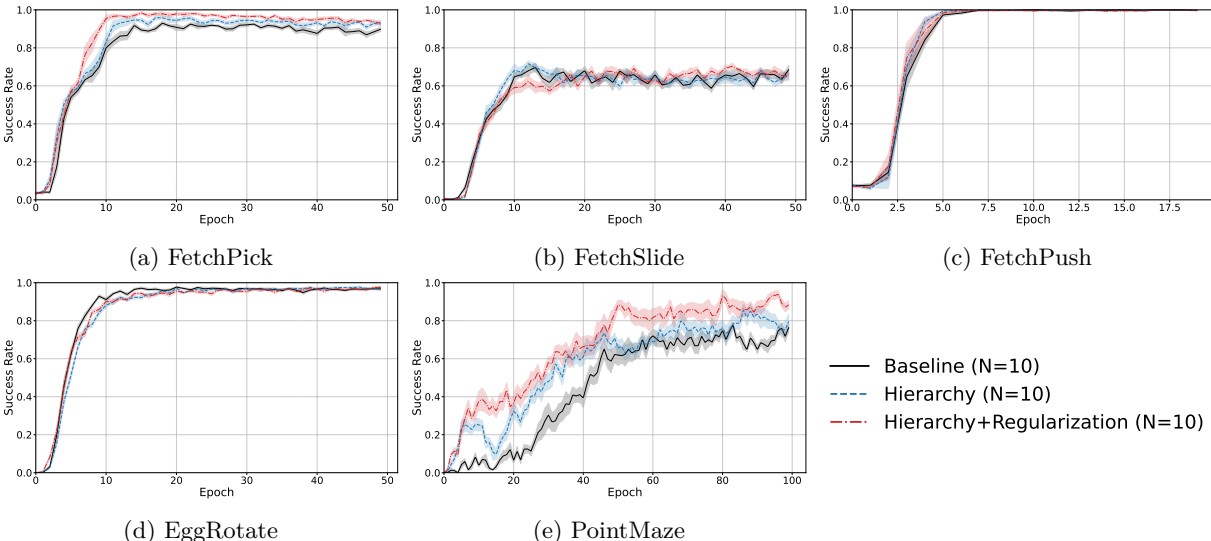

Figure 4: Success rate for the Capable Low-level Policy setting. The solid lines represent the mean success rate over 10 random seeds, and the shaded regions indicate the standard error.

### 5.3.2 Performance with a Capable Low-level Policy

We next evaluate the capable-worker setting as a complementary check to verify that the proposed hierarchical framework remains competitive when the low-level policy is sufficiently expressive. The corresponding learning curves are shown in Figure 4.

Across the suite of robotic manipulation tasks (FetchPick, FetchSlide, FetchPush, and EggRotate), all three approaches demonstrated strong results. These results confirm that when the low-level controller is sufficiently expressive, our decoupled hierarchical structure remains competitive with a highly capable non-hierarchical agent.

The main difference in this setting appears in the learning curves on PointMaze. Unlike the manipulation tasks, PointMaze requires planning around obstacles, so success depends more strongly on capturing the

global structure of feasible paths rather than only local reachability. In this task, the hierarchical variants tend to improve more quickly than the non-hierarchical baseline during the early phase of learning, and **Hierarchy+Regularization** tends to achieve the best performance overall. This pattern is consistent with the idea that a more geometrically informed high-level representation can be particularly useful for subgoal selection in obstacle-rich navigation. Overall, these results suggest that, in the capable low-level setting, the proposed hierarchy remains competitive across tasks, while the regularized variant may provide an additional benefit in environments with more complex global structure.

### 5.4   Robustness with Multiple Specialized Low-Level Controllers

To further illustrate a modular extension of our decoupled framework, we consider a scenario that mimics a more advanced edge-computing setup involving multiple specialized control modules. This experiment tests the hypothesis that an ensemble of simple, specialized low-level actors can be more effective and parameter-efficient than a single, monolithic, high-capacity actor.

**Experimental Setup.**   We use the PointMaze environment for this experiment. The state space is partitioned into four regions based on the XY coordinates (quadrants). A dedicated, resource-constrained low-level actor, $\pi_{\mathrm{ll}}^{(i)}$ (with 2 layers and 8 hidden units), is assigned to each quadrant $i \in \{1, 2, 3, 4\}$. During execution and training, the low-level actor $\pi_{\mathrm{ll}}^{(i)}$ is assigned based on the agent's current state $s_t$. The high-level planning architecture remains unchanged from the previous experiments. The performance of this multi-actor setup is compared against the **Single Capable** policy (4 layers, 256 hidden units), i.e., the same non-hierarchical baseline as in Sec. 5.3.2. Notably, the total number of parameters for the four specialist actors is smaller than that of the single capable baseline actor.

**Results and Discussion.**   As shown in Figure 5, our findings reveal two key insights. First, our hierarchical framework using the four low-capacity actors without trajectory regularization achieved a final success rate comparable to that of the single, higher-capacity, capable actor baseline. This suggests that by partitioning the control problem, our method can achieve the same level of performance with better parameter efficiency. This is consistent with the hypothesis that for a smaller, specialized portion of the state space, a low-capacity network is sufficient to learn an effective control policy.

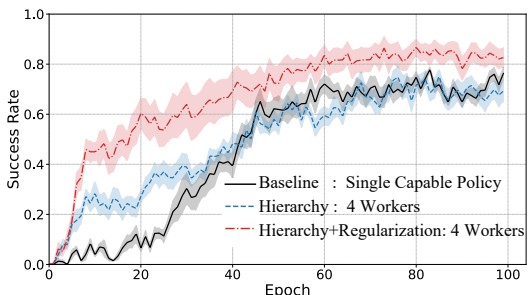

Figure 5: Mean success rate in PointMaze using 4 specialized low-capacity workers with standard error over 10 random seeds.

Additionally, when trajectory regularization was incorporated, the multi-actor setup performed even better. It learned faster in the early stage and generally achieved the highest success rate over the course of training. These results suggest a useful interaction within our framework:

1. The trajectory regularization contributes to a more accurate global "map" for high-level planning.

2. This improved map supports more precise and effective subgoal generation.

3. As a result, effective subgoal guidance allows an ensemble of simple, parameter-efficient, specialized actors to match or outperform a single, complex, generalist policy.

This suggests that the global planning capability of our hierarchical framework can effectively compensate for and leverage the simplicity of localized, low-capacity controllers, supporting its potential usefulness in modular and resource-constrained systems.

# 6    Conclusions

This paper introduces a metric-based Hierarchical Reinforcement Learning (HRL) framework that mitigates a major source of planner instability by **decoupling** high-level planning from low-level control. The central idea is to reframe the high-level policy as a planner operating on a learned **metric space**—a "map" of the environment's structure captured by the value function. This avoids training the planner via RL on non-stationary transitions induced by the evolving low-level policy. By architecturally enforcing the triangle inequality in the critic via a Metric Residual Network (MRN), our framework learns a geometric map on which planning is simplified to identifying subgoals along geodesics. The proposed trajectory regularization can further encourage geometric consistency in this map.

This decoupled approach demonstrates strong robustness. Our experiments show that while our framework performs comparably to strong non-hierarchical baselines with a capable low-level controller, it outperforms in **resource-constrained scenarios** where standard baselines fail, and representative HRL methods become substantially less effective (see Appendix B). This finding is particularly relevant for real-world applications such as edge-cloud robotics, where on-device controllers are often limited. Furthermore, we demonstrate that our planner can effectively coordinate an ensemble of simple, specialized low-level actors, achieving superior performance with better parameter efficiency than a single monolithic policy. This highlights the broader applicability of our design.

In summary, by interpreting the value function as a distance metric, our framework enables geometry-aware planning that is robust to limitations in low-level controllers. This work represents a key step toward bridging the gap between high-level AI planning and practical, resource-constrained robotics.

### Acknowledgments

This work was supported by JST SPRING, Grant Number JPMJSP2110 (to S.M.) and JSPS KAKENHI (22H04998, 23H04676 and 25K24537 to S.I.).

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

# A    Multi-Step Planning via Iterative Subgoal Refinement

The core framework described in the main text enables the high-level policy, $\pi_{\text{hl}}$, to generate a single intermediate subgoal. However, for long-horizon tasks, this subgoal, while guaranteed to lie on the geodesic, might be so distant from the current state that it offers little advantage over pursuing the final goal directly.

To address this, we generalize the framework to an $N$-step planning process, termed **iterative subgoal refinement**. This technique recursively leverages the same trained high-level policy, $\pi_{\text{hl}}$, to decompose the path into a sequence of $N$ waypoints, thereby yielding a more proximal and readily achievable subgoal for the low-level policy to execute. This section details the training and execution procedures for the general $N$-hop case. The experiments in this paper consistently use a 2-hop ($N = 2$) refinement process.

## A.1    Training Objective for Iterative Refinement

To enable iterative refinement, the high-level policy must be trained not only to find a single waypoint but also to decompose a long trajectory into a sequence of shorter segments that lie on the geodesic path.

Let $N$ be the total number of refinement hops. Given the latent representation of the current **state-action context**, $z_1 = e_1(s, a)$, and the final **goal context**, $z_g = e_2(s, g)$, we define the final goal as the initial waypoint $z_{\text{w}}^{(0)} = z_g$. We then generate a sequence of $N$ subsequent latent waypoints, $\{z_{\text{w}}^{(i)}\}_{i=1}^{N}$, in a recursive manner. This generation process iteratively refines the waypoint, starting from the final goal and moving closer to the current state:

$$z_{\text{w}}^{(1)} = \pi_{\text{hl}}(z_1, z_{\text{w}}^{(0)}) \tag{5}$$

$$z_{\text{w}}^{(2)} = \pi_{\text{hl}}(z_1, z_{\text{w}}^{(1)}) \tag{6}$$

$$\vdots$$

$$z_{\text{w}}^{(N)} = \pi_{\text{hl}}(z_1, z_{\text{w}}^{(N-1)}) \tag{7}$$

As conceptually illustrated in Figure 6, this process begins by generating the first waypoint, $z_{\text{w}}^{(1)}$, along the geodesic path from the current state context $z_1$ to the final goal $z_{\text{w}}^{(0)}(= z_g)$. Each subsequent waypoint $z_{\text{w}}^{(i)}$ is a refinement of the previously generated waypoint $z_{\text{w}}^{(i-1)}$. The final waypoint, $z_{\text{w}}^{(N)}$, is therefore expected to represent the most immediate subgoal for the low-level policy.

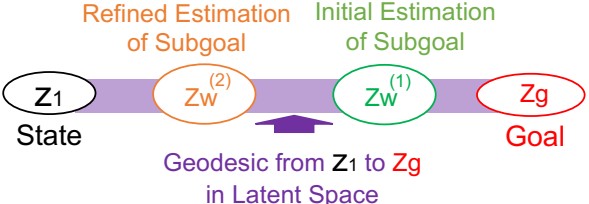

Figure 6: Conceptual diagram of the 2-hop subgoal estimation in latent space. The initial waypoint $z_w^{(1)}$ is estimated from $z_1$ and $z_g$. The 2nd refined subgoal is then generated from $z_1$ and the initial estimation $z_w^{(1)}$.

The high-level policy is trained to find a sequence of waypoints that lie on the geodesic path by maximizing the cumulative value along the entire multi-segment trajectory. The generalized objective function for $N$-hop planning is therefore defined as:

$$J_{\text{hl}}(\theta; s, a, g) = Q_\theta(z_1, z_{\text{w}}^{(N)}) + \sum_{i=1}^{N} Q_\theta(z_{\text{w}}^{(i)}, z_{\text{w}}^{(i-1)}) \tag{8}$$

To train the multi-hop planner, this generalized objective replaces the single-step version (Equation 1) in our main training procedure (Algorithm 1). By maximizing it, the planner learns to find a sequence of waypoints

along the geodesic, pushing the path's value toward its theoretical upper bound of $Q_\theta(z_1, z_g)$, as formally derived in Appendix A.5.

## A.2 Execution-Time Subgoal Generation

During execution (for both data collection and evaluation), the agent uses the trained policy $\pi_{\mathrm{hl}}$ to determine the most immediate subgoal. The process, detailed in Algorithm 3 and conceptually illustrated for a 2-hop case in Figure 7, starts with the final goal and iteratively refines it to find a subgoal that is readily achievable from the current state.

The process begins by encoding the final goal into a latent waypoint, $z_{\mathrm{w}} \leftarrow e_2(s_t, g_{\mathrm{final}})$. The policy $\pi_{\mathrm{hl}}$ is then applied sequentially for $N_{\mathrm{hops}}$ iterations. In each iteration, the policy refines the current waypoint $z_{\mathrm{w}}$ into a new, closer one, using a state-action context $z_1$ that is re-estimated at each step. After $N_{\mathrm{hops}}$ iterations, the resulting latent waypoint is decoded into $\tilde{g}_{\mathrm{subgoal}}$ and passed to the low-level policy.

---

**Algorithm 3** Action Selection with Iterative Subgoal Refinement

---

1: **Input:** Current state $s_t$, final goal $g_{\mathrm{final}}$, number of refinement hops $N_{\mathrm{hops}}$.
2: **Output:** Action $a_t$ to be executed.

3: — *Plan an immediate subgoal via iterative refinement* —
4: $z_{\mathrm{w}} \leftarrow e_2(s_t, g_{\mathrm{final}})$            ▷ Initialize waypoint as the final goal $(z_{\mathrm{w}}^{(0)})$
5: **for** $i = 1$ **to** $N_{\mathrm{hops}}$ **do**
6:    $g_{\mathrm{w}} \leftarrow D_\psi(z_{\mathrm{w}})$               ▷ Decode current waypoint
7:    $a_{\mathrm{est}} \leftarrow \pi_{\mathrm{ll}}(s_t, g_{\mathrm{w}})$         ▷ Estimate action using closer waypoint
8:    $z_1 \leftarrow e_1(s_t, a_{\mathrm{est}})$           ▷ Encode state-action context
9:    $z_{\mathrm{w}} \leftarrow \pi_{\mathrm{hl}}(z_1, z_{\mathrm{w}})$         ▷ Refine waypoint to a closer one
10: **end for**

11: — *Execute action based on the final refined subgoal* —
12: $\tilde{g}_{\mathrm{subgoal}} \leftarrow D_\psi(z_{\mathrm{w}})$          ▷ Decode the final waypoint $z_{\mathrm{w}}^{(N)}$
13: $a_t \leftarrow \pi_{\mathrm{ll}}(s_t, \tilde{g}_{\mathrm{subgoal}})$        ▷ Generate action to reach the subgoal
14: **return** $a_t$

---

**Implementation notes** For notational simplicity, Equations (5)~(7) show a single state-action context, $z_1$. In practice, we dynamically update this context at each refinement step. To generate waypoint $z_{\mathrm{w}}^{(i)}$, we compute a new context $z_1$ using an action $a_{\mathrm{est}}$ estimated toward the previous waypoint, $z_{\mathrm{w}}^{(i-1)}$. This provides the planner with a more accurate context based on a closer, more relevant target.

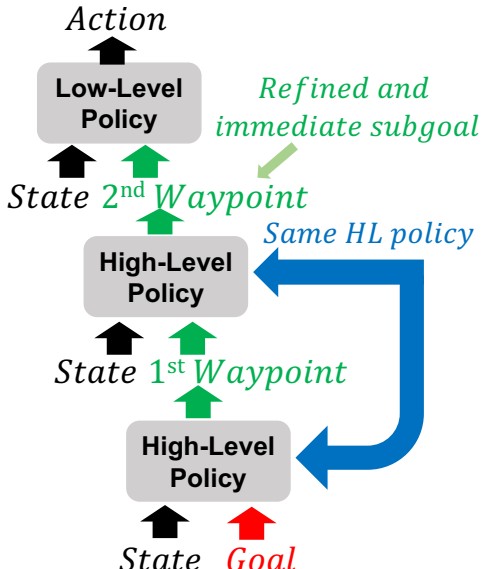

Figure 7: Conceptual diagram of 2 hop Hierarchical RL

### A.3 Empirical Analysis of Refinement Hops

This section provides an empirical analysis to determine the optimal number of refinement hops, $N$, for the iterative subgoal refinement process detailed in Appendix A.1 and A.2. We compared the performance of our hierarchical framework by varying the number of high-level policy applications among 1-hop, 2-hop, and 4-hop settings. These experiments were conducted under the *Resource-Constrained Low-level Policy* scenario, where the robustness of our method is most critical. The results are presented in Figure 8.

Our findings indicate that for most tasks—specifically FetchPick, FetchSlide, FetchPush, and PointMaze—there was no significant performance difference among the tested hop counts. However, a notable exception was observed in the EggRotate task, where the 2-hop configuration clearly outperformed the 1-hop setup. We hypothesize that this distinction is linked to the high dimensionality of the task's state space. As shown in Table 1, the state dimension of EggRotate (61) is considerably larger than that of the other environments (Fetch tasks: 25, PointMaze: 4). In tasks with such complex, high-dimensional state spaces, a multi-step planning approach like 2-hops, which decomposes the path to the final goal into finer intermediate subgoals, appears to facilitate more effective learning.

Conversely, the 4-hop setting resulted in a significant performance degradation on the EggRotate task, suggesting that an excessive number of hops can be counterproductive. The likely cause is training instability. As $N$ increases, the objective function for multi-hop planning becomes substantially more complex. The longer chain of dependencies in the 4-hop objective function can introduce optimization challenges and numerical instability, preventing the high-level policy from converging to an effective strategy and thereby degrading the overall learning process.

Based on this analysis, we adopted a 2-hop ($N = 2$) configuration for all main experiments presented in this paper. We concluded that this setting offers the best balance between generality and performance, as it demonstrated a clear benefit in the most complex task without compromising performance in simpler ones.

### A.4 Qualitative Analysis of Subgoal Generation

Figure 9 provides a qualitative visualization of the subgoal generation process in the PointMaze environment, shown as a sequence of three snapshots from a single episode. This example illustrates the practical success of our full method (Hierarchy+Regularization), whose quantitative performance is shown in Figure 4e.

Table 1: The state dimension of different environments

| FetchPick | FetchSlide | FetchPush | EggRotate | PointMaze |
|-----------|------------|-----------|-----------|-----------|
| 25 | 25 | 25 | 61 | 4 |

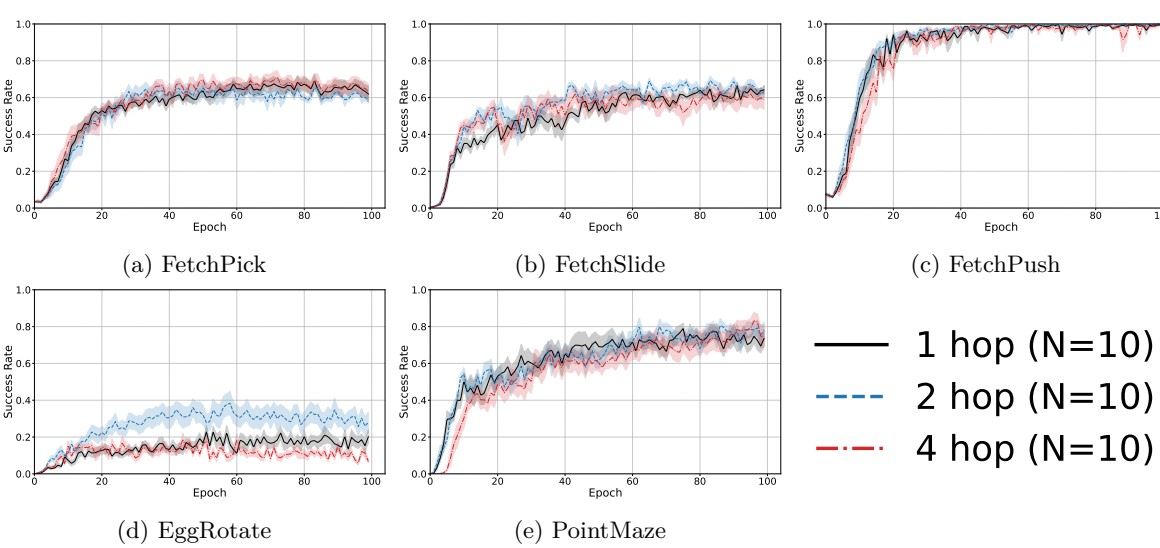

(a) FetchPick  (b) FetchSlide  (c) FetchPush

(d) EggRotate  (e) PointMaze

Figure 8: Success rate for different numbers of high-level planning hops in the Resource-Constrained Low-level Policy setting. The solid lines represent the mean success rate over 10 random seeds, and the shaded regions indicate the standard error.

The key observation is that the high-level planner successfully generates two intermediate waypoints that lie on the **geodesic path**—the true shortest path that navigates the maze's topology. This demonstrates that the learned metric space has captured the non-Euclidean structure of the environment, avoiding naive straight-line paths that would collide with walls.

Furthermore, the figure highlights the iterative refinement process. Waypoint 1 is the initial, more distant subgoal proposed based on the current state and final goal. As shown in the snapshots, Waypoint 2 is a subsequent refinement, generated using the current state and Waypoint 1 as inputs. As a result, Waypoint 2 is positioned closer to the agent, serving as a more immediate and effective target for the low-level policy.

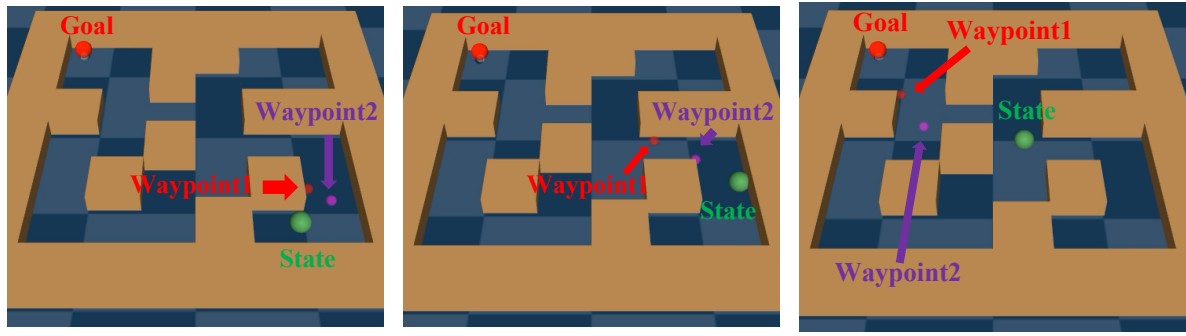

Figure 9: A sequence of snapshots from a single episode illustrating the 2-hop iterative refinement in the PointMaze environment. The first waypoint is generated from the final goal. The second waypoint is a refinement of the first, representing a closer, immediate subgoal for the low-level actor (green ball). The low-level actor pursues this second waypoint.

### A.5 Derivation and Upper Bound of The Multi-Hop Objective Function

The design of our high-level planner's objective function is rooted in the geometric properties of the critic's value function, $Q_\theta$. By construction, the Metric Residual Network (MRN) ensures that the negative of the critic, $-Q_\theta$, functions as a **quasipseudometric**, which by definition satisfies the triangle inequality. This implies a key property for the value function itself. For any three latent points $z_i, z_j, z_k$ in the latent space, the following inequality holds:

$$Q_\theta(z_i, z_j) + Q_\theta(z_j, z_k) \leq Q_\theta(z_i, z_k) \tag{9}$$

This inequality means that the value accumulated along any two-step path is, at most, the value of taking the direct path. The planner's objective is to find an intermediate point $z_j$ that lies on the geodesic (shortest path), which turns this inequality into an equality.

The multi-hop objective is derived by recursively applying this fundamental inequality (Equation 9). Let $z_1$ be the current state-action context and $z_g$ be the final goal context, which we denote as the initial waypoint $z_w^{(0)} = z_g$. The planner, $\pi_{hl}$, generates a sequence of waypoints $\{z_w^{(i)}\}_{i=1}^N$.

By applying Equation 9 repeatedly, we can establish an upper bound for the cumulative value of a multi-segment path. For a 2-hop path ($N = 2$), for instance:

$$\underbrace{\left(Q_\theta(z_1, z_w^{(2)}) + Q_\theta(z_w^{(2)}, z_w^{(1)})\right)}_{\leq Q_\theta(z_1, z_w^{(1)}) \text{ by Eq. (9)}} + Q_\theta(z_w^{(1)}, z_w^{(0)})$$

$$\leq \underbrace{Q_\theta(z_1, z_w^{(1)}) + Q_\theta(z_w^{(1)}, z_w^{(0)})}_{\leq Q_\theta(z_1, z_w^{(0)}) \text{ by Eq. (9)}}$$

$$\leq Q_\theta(z_1, z_w^{(0)}) = Q_\theta(z_1, z_g)$$

Generalizing this by induction, the total value of the N-hop path is upper-bounded by the value of the direct path from the current state to the final goal:

$$Q_\theta(z_1, z_w^{(N)}) + \sum_{i=1}^N Q_\theta(z_w^{(i)}, z_w^{(i-1)}) \leq Q_\theta(z_1, z_g)$$

Our high-level policy is trained to maximize the left-hand side of this inequality. This is precisely the N-hop objective function (Equation 8):

$$J_{hl}^{(N)} = Q_\theta(z_1, z_w^{(N)}) + \sum_{i=1}^N Q_\theta(z_w^{(i)}, z_w^{(i-1)})$$

By maximizing this objective, the planner is trained to select a sequence of waypoints that brings the path's total value as close as possible to its theoretical upper bound, $Q_\theta(z_1, z_g)$. Consequently, the generated sequence of subgoals is encouraged to lie along the optimal, geodesic path within the learned metric space.

## B Failure Analysis of Tightly-Coupled HRL in Resource-Constrained Settings

A natural question is how our proposed method compares to existing Hierarchical Reinforcement Learning (HRL) frameworks, such as HIRO (Nachum et al., 2018). While a direct one-to-one comparison is valuable, it is challenging in this case due to fundamental differences in their underlying components and assumptions. Our framework is built upon a Metric Residual Network (MRN) critic, which is architecturally designed to learn a metric space and operates in conjunction with Hindsight Experience Replay (HER) and negative sparse rewards. Rather than modifying HIRO to match these design choices, we focus on a more direct question: whether a representative tightly-coupled HRL method remains robust when the low-level worker is severely resource-constrained.

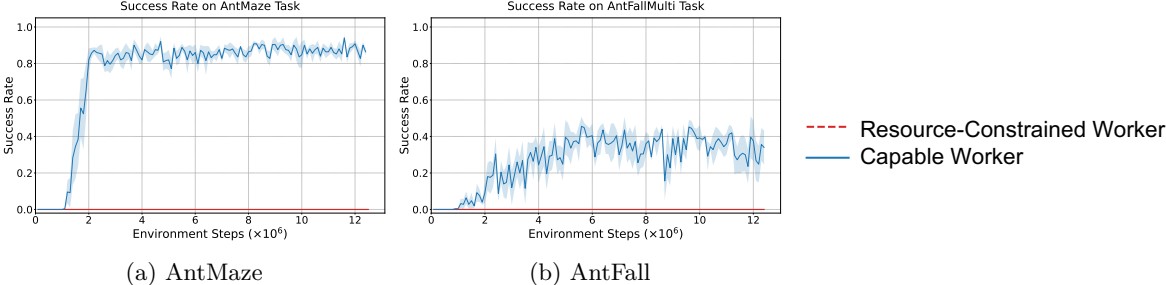

(a) AntMaze        (b) AntFall

Figure 10: Success rate of the HIRO agent on AntMaze and AntFall tasks. We compare a **Capable Worker** (high-capacity) against a **Resource-Constrained Worker** (low-capacity). The resource-constrained worker completely fails to learn, highlighting the brittleness of conventional tightly-coupled HRL frameworks to worker capacity limitations. The solid lines represent the mean success rate over 5 random seeds, and the shaded regions indicate the standard error.

To test this, we evaluate HIRO under the same low-capacity setting considered in our main experiments. Specifically, we compare the performance of HIRO with two different low-level worker architectures:

- **Capable Worker**: A high-capacity network with two 300-unit hidden layers, as specified in the original HIRO paper.

- **Resource-Constrained Worker**: A low-capacity network with two 8-unit hidden layers, mirroring the constrained setting from our main paper.

The results are presented in Figure 10. In both the AntMaze and AntFall environments, which were used in the original HIRO benchmark, the agent with a **Capable Worker** is able to learn successfully. However, the agent with a **Resource-Constrained Worker** fails completely, making no meaningful progress on the tasks. This outcome demonstrates that the performance of tightly-coupled HRL methods like HIRO is highly sensitive to the capacity of the low-level worker. The failure of the resource-constrained agent validates the core motivation for our work: that a new approach is needed for real-world scenarios where low-level controllers are inherently limited.

## C Environment and Task Details

To evaluate the generality and robustness of our proposed framework, we selected a suite of goal-conditioned environments from the Gymnasium-Robotics library, covering both navigation and robotic manipulation (Plappert et al., 2018; Fu et al., 2020). As highlighted in the main text, these tasks were chosen to impose qualitatively different challenges on the agent, ranging from topological complexity to high-dimensional control. All tasks use the standard sparse reward function: $r_t = 0$ if the object is within the threshold $\epsilon$ of the goal, and $r_t = -1$ otherwise.

### C.1 Navigation Task: PointMaze

- **Environment:** `PointMaze_Medium_Diverse_GR-v3`

- **Challenge (Topological Complexity):** This task requires a point-mass agent to navigate a maze to reach target coordinates. The "Medium" maze layout includes multiple walls and obstacles, creating a complex non-Euclidean topology. This setup evaluates whether the learned metric space correctly captures the geodesic distance (the actual shortest path avoiding walls) rather than the misleading Euclidean distance.

- **State Space ($\mathcal{S}$):** 4-dimensional continuous space consisting of the agent's position $(x, y)$ and velocity $(v_x, v_y)$.

- **Goal Space ($\mathcal{G}$):** 2-dimensional space representing the target coordinates $(x_g, y_g)$.

- **Action Space ($\mathcal{A}$):** 2-dimensional continuous action corresponding to the *linear force* applied to the point-mass.

## C.2   Robotic Manipulation Tasks: Fetch

- **Environment:** `FetchPickAndPlace-v4 / FetchPush-v4 / FetchSlide-v4`

- **Challenge (Diverse dynamics):** These Fetch tasks span a range of manipulation dynamics, including grasp-and-transport (PICKANDPLACE), contact-rich pushing (PUSH), and low-friction sliding (SLIDE). In particular, SLIDE requires precise impulse control to send the puck *sliding* toward a distant goal, making friction and inertia the dominant factors for success.

- **State Space ($\mathcal{S}$):** 25-dimensional continuous vector including gripper and object kinematics (positions, velocities, orientations) and relative features.

- **Goal Space ($\mathcal{G}$):** 3-dimensional target position.

- **Action Space ($\mathcal{A}$):** 4-dimensional continuous action controlling Cartesian displacement $(dx, dy, dz)$ and a continuous gripper open/close signal.

## C.3   Dexterous Manipulation Task: Shadow Hand

- **Environment:** `HandManipulateEggRotate-v1`

- **Challenge (High Dimensionality):** This task involves controlling a 24-DOF Shadow Hand to rotate an egg-shaped object to a specific target orientation.

- **State Space ($\mathcal{S}$):** 61-dimensional vector consisting of the hand joint states and the egg state (including its position and quaternion orientation).

- **Goal Space ($\mathcal{G}$):** 7-dimensional vector representing the target position and orientation (quaternion) of the object.

- **Action Space ($\mathcal{A}$):** 20-dimensional vector specifying absolute angular positions of the actuated joints.

# D   Control Experiment: Freezing the Low-Level System in HIRO

To test whether the gain of our method can be explained by freezing the low-level system alone, we implemented an additional control based on HIRO, denoted as **HIRO + Freeze**. This variant first pre-trains the low-level actor and critic, and then freezes both while continuing high-level learning. We evaluate this control under the same dense-reward setting as Appendix B.

Figure 11 compares standard HIRO and **HIRO + Freeze** with both resource-constrained and capable workers. The results do not support the explanation that our gain comes simply from freezing. In the **resource-constrained** setting, both variants show no meaningful learning, with success remaining at zero across all tested environments. Thus, simply fixing the low-level system is not sufficient to recover effective hierarchical learning in the regime that motivates our method.

In the **capable-worker** setting, both variants are able to learn, but freezing does not provide a consistent benefit. On AntMaze, the frozen variant shows at most a small early advantage, likely due to the additional low-level pre-training, but this does not translate into a clear final improvement. On AntFall, we do not observe a meaningful advantage from freezing, and performance is similar to or slightly worse than standard HIRO.

Overall, these results show that the gain of our method cannot be explained by freezing alone. Even with a fixed low-level actor and critic, HIRO still learns the high-level policy from worker-induced high-level

transitions. Under resource-constrained workers, these transitions remain poor, so freezing alone does not recover effective planning. By contrast, our method optimizes the planner against a critic-defined geometric objective instead of high-level RL on worker-induced dynamics.

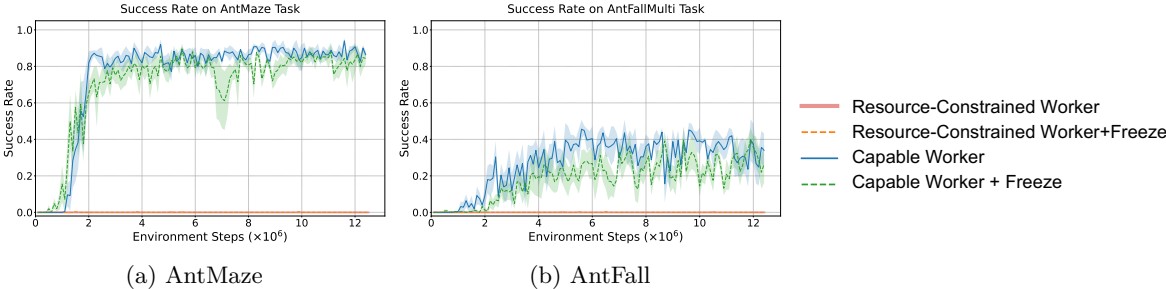

| (a) AntMaze | (b) AntFall |

Figure 11: Success rate of standard HIRO and **HIRO + Freeze** under resource-constrained and capable workers on AntMaze and AntFall. **HIRO + Freeze** first pre-trains the low-level actor and critic, and then freezes both during high-level learning. Under resource-constrained workers, both variants fail across the tested environments, whereas under capable workers both can learn but freezing provides no consistent final benefit. Solid lines show the mean over 5 random seeds, and shaded regions indicate the standard error.

## E Supplementary Three-Way Comparison under a Shared Sparse Goal-Conditioned Reward Setting

To complement the within-HIRO control in Appendix D, we report a supplementary three-way comparison among **our full proposed method**, **HIRO**, and **HIRO + Freeze** under a shared sparse goal-conditioned reward setting in the capable-worker regime. The purpose of this experiment is to provide a single cross-method view under a common setting. This comparison should be interpreted as supplementary rather than as a primary fairness-matched benchmark, since the methods were not originally introduced under an identical training setup.

Appendix D follows the reward and training regime closest to the original HIRO formulation in order to isolate the freezing intervention within that framework. By contrast, the present section places all methods in our sparse goal-conditioned setting to provide a shared cross-method view.

We evaluate all three methods on the same environments and with the same success-rate metric used in Section 5. Here, **HIRO + Freeze** denotes the same intervention studied in Appendix D: the low-level actor and critic are first pre-trained and then fixed for the remainder of training, while the high-level policy continues to be updated. By contrast, our method is the same as in the main text: the critic and low-level policy are updated throughout training and are frozen only temporarily during each high-level update.

The results are shown in Figure 12. Under this sparse setting, neither **HIRO** nor **HIRO + Freeze** shows reliable learning. In most environments, both remain near zero success throughout training. In PointMaze, **HIRO + Freeze** starts from a small nonzero success level, but shows little improvement over training, while **HIRO** stays at a slightly lower success level overall. This suggests that pre-training may provide limited initial competence in this environment, but **permanent freezing alone does not recover effective learning**. By contrast, **our method** shows clear learning progress under the same setting.

This shared-setting comparison complements Appendix D, but does not replace the control reported there. Appendix D isolates the freezing intervention under a setting close to the original HIRO formulation, whereas the present section provides a shared sparse-setting comparison across methods. Taken together, these results support the conclusion that the gains of our method cannot be explained by freezing alone, but instead arise from optimizing the planner against a critic-defined geometric objective.

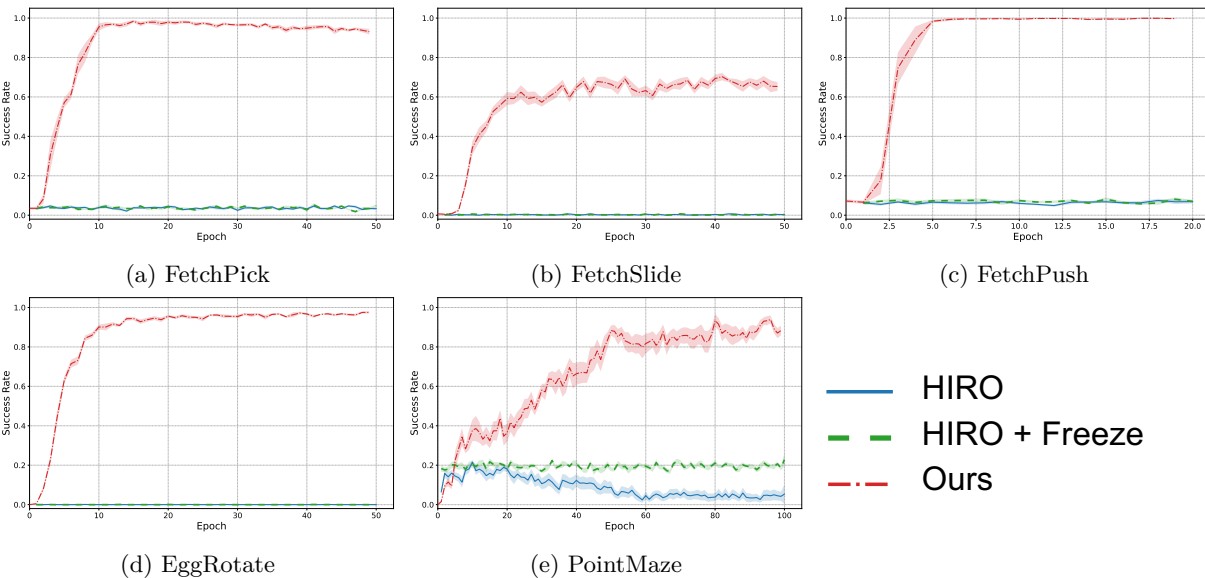

Figure 12: Success rate under a sparse goal-conditioned reward setting for **HIRO**, **HIRO + Freeze**, and **our full proposed method**. Under this setting, **our method** shows clear learning progress, whereas both HIRO variants fail to learn reliably. In most environments, their success remains near zero throughout training; in PointMaze, **HIRO + Freeze** reaches a small nonzero success rate but shows little improvement over training. Solid lines show mean success over 10 random seeds, and shaded regions indicate the standard error.

# F Impact of the Explicit Metric Structure on Hierarchical Planning

To evaluate the impact of the explicit metric structure on our planner, we performed an additional ablation in which we replace the MRN distance computation while keeping the rest of the framework as unchanged as possible. The goal is to separate the effect of our hierarchical planning objective from the effect of using a critic with an explicit geometric structure as the planning substrate.

In our full model, the critic is built from latent representations

$$z_1 = e_1(s, a), \qquad z_2 = e_2(s, g),$$

followed by the MRN distance decomposition

$$d_{\mathrm{MRN}}(z_1, z_2) = d_{\mathrm{sym}}(z_1, z_2) + d_{\mathrm{asym}}(z_1, z_2),$$

where the symmetric and asymmetric terms impose an explicit quasimetric structure. The critic is then given by

$$Q_\theta(s, a, g) = -d_{\mathrm{MRN}}(z_1, z_2).$$

In this ablation, we keep the same latent encoders $e_1, e_2$, but replace the above distance computation with a learned scalar function on the concatenated latent pair:

$$Q_\psi^{\mathrm{concat}}(s, a, g) = -\rho(f_\psi([z_1; z_2])), \qquad (10)$$

where $f_\psi$ is a two-layer MLP and $\rho(x) = \log(1 + \exp(x))$ is the softplus function, used to enforce nonnegativity of the output. This keeps the critic non-positive, but removes the explicit symmetric/asymmetric decomposition and hence the explicit triangle-inequality structure in the latent space.

We evaluate this critic variant in two settings: (i) the standard non-hierarchical goal-conditioned baseline, and (ii) our hierarchical framework, in which the high-level planning objective uses this critic in place of the

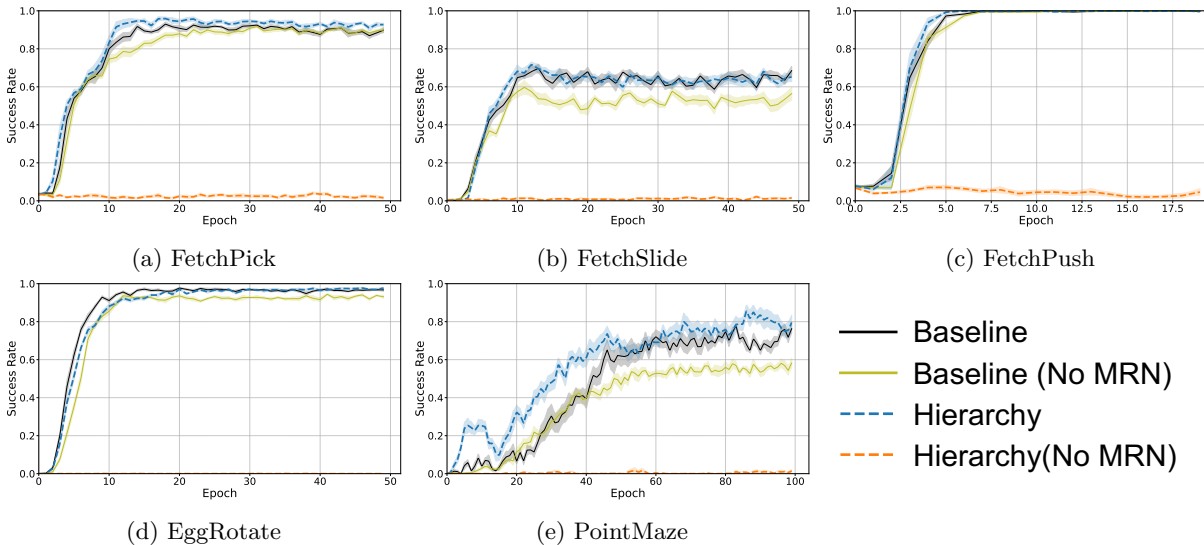

(a) FetchPick        (b) FetchSlide        (c) FetchPush

(d) EggRotate        (e) PointMaze

Figure 13: Success rate for the ablation in which the explicit MRN-based metric structure is replaced with a simple concatenation-based critic. In the standard non-hierarchical baseline, this critic still supports learning in several environments, although it is generally less sample-efficient than the original MRN-based baseline and often reaches lower final performance. In contrast, when the same critic is used in the hierarchical planner, performance collapses in most environments, with success remaining near zero throughout training. Solid lines show the mean success rate over 10 random seeds, and shaded regions indicate the standard error.

original MRN-based critic. Thus, the only change in this ablation is that the explicit quasimetric structure is replaced with a learned scalar function of the latent pair.

The results show a clear asymmetry. In the standard non-hierarchical baseline, the concatenation-based critic (**Baseline (No MRN)**) still supports learning in several environments, although it is generally less sample-efficient than the original MRN-based baseline and sometimes reaches lower final performance. In contrast, when the same critic replacement is applied in our hierarchical framework (**Hierarchy (No MRN)**), performance degrades sharply: in most environments, the success rate remains near zero throughout training (Figure 13).

This difference is consistent with the role of MRN in our method. In the flat baseline, the critic only needs to provide a useful objective for learning the goal-conditioned policy. In the hierarchical setting, however, the critic is used as a *planning map*: the planner relies on the critic to represent a geometry of travel cost over intermediate subgoals, and in our method this role is supported by the explicit metric structure of MRN. Once this geometric structure is removed, the same planner objective no longer provides a reliable basis for subgoal selection. Overall, these results show that hierarchy alone is not sufficient; the explicit metric structure is important specifically for hierarchical planning.

