# OpenReview forum: "Decoupling Planning from Control: Stable Hierarchical RL with a Learned Metric Space"
_TMLR — Accepted by TMLR_

### Review · Reviewer_VqMh · 2026-02-03

**Summary Of Contributions:**

The authors propose a hierarchical reinforcement learning framework that decouples high-level planning from low-level control, such that it addresses the non-stationary problem in hierarchical RL, and stabilizes learning. They also propose a novel trajectory regularization loss that enforces geometric consistency along the agent’s trajectories. The authors show that in scenarios where the low-level policy is resource-constrained, their proposed method works well where standard approaches fail.

**Audience:**

Yes

**Audience Explanation:**

This paper would be of interest for individuals interested in RL and hierarchical RL.

**Claims And Evidence:**

No

**Claims Explanation:**

The authors make three claims in this paper:

1) That their proposed hierarchical reinforcement learning framework effectively decouples high-level planning from low-level control, such that it addresses the non-stationary problem in hierarchical RL, and stabilizes learning.
2) That their proposed trajectory regularization loss enforces geometric consistency along the agent’s trajectories.
3) That in scenarios where the low-level policy is resource-constrained, their proposed method works well where standard approaches fail.

In terms of the first claim, the authors claim that their proposed method addresses the non-stationarity problem. Yet, one could argue that the biggest contributing factor that may yield such a result is the choice to freeze the lower-level policy/critic during the training of the high-level policy (see Section 3.3). This is something that could also be done with a standard HRL approach, however the authors do not make such a comparison. As such, it is unclear the extent to which their proposed method addresses the non-stationarity problem vs. the training procedure of freezing the low-level system. Accordingly, although the authors do decouple the high-level planning from low-level control, I do not find the claim regarding non-stationarity adequately supported.

In terms of the second claim, the authors derive a loss that penalizes deviations from the triangle inequality. As such, if one presumes that adhering to the triangle inequality enforces geometric consistency along the agent’s trajectories, then the claim is justified. However, Section 3.1, which argues the above presumption, lacks rigorous justification. More specifically,  the claim that “This principle is mathematically captured by the triangle inequality” is not proved by the authors or supported by a citation. Accordingly, at the moment I do not find the second claim adequately supported.

In terms of the third claim, I find that it is adequately supported by the empirical evidence presented in Section 4.3.2.

**Requested Changes:**

From a writing perspective, the paper is generally well-written and I have no major concerns in that regard. My only note is that all figures should be referenced in the main body text; at the moment, Figure 1 is never referenced in the main text. The notation is generally easy to follow, but a bit sloppy at times. For example, the reward of the GCMDP (Section 2.1) should be explicitly defined as a scalar (i.e., $R \in \mathbb{R}$). Similarly, the transition dynamics should be more rigorously defined.

In terms of the figures, their quality is at a minimally-acceptable level, however, I challenge the authors to improve the quality of the figures. At the moment, they (Figures 1, 2) resemble figures that would go into a schoolwork report rather than a research publication.

Major concerns/requests are as follows:
- I think the authors could do a better job in motivating why a subgoal approach is needed in the first place. In particular, why not just learn a model that goes from any given state to the goal state, rather than some intermediate subgoal? Articulating this argument will make the paper, and the proposed method, more compelling.
- Section 3.1: the claim that “This principle is mathematically captured by the triangle inequality” needs to be proved by the authors or supported by a citation.
- To properly gauge the effectiveness of the proposed method as it relates to the non-stationary claims, the authors need to compare their method to a HRL baseline that also freezes the low-level system.

Other requests are as follows:
- Section 2.4: “MLP” is never defined.
- Section 2.4: (i) claims that the decomposition is sample efficient, however this is not justified by either citing prior work that shows it, or a more rigorous argument/justification by the authors.
- Section 2.4: (ii) needs to be better motivated (i.e., why are we talking about it in the first place?). Moreover, it is not clear whether the contents being presented here are novel or a summary of prior work. If it is based on prior work, a citation is needed, otherwise, the authors should be explicit that this is part of their contribution.
- Section 3.1: moving the relevant pseudocode to this section might improve the readability of the section.

---

> ### Author Response · Authors · 2026-03-11
> **Author Response**
>
> We thank the reviewer for the careful reading and constructive feedback. We agree that the original manuscript was not sufficiently clear about (i) the scope of our non-stationarity claim, (ii) the justification for the triangle-inequality argument, and (iii) the motivation and positioning of several technical components. We revised the paper accordingly.
>
> **(1) To properly support the non-stationarity claim, the paper should compare against an HRL baseline that also freezes the low-level system.**
>
> We agree and added the requested control experiment in **Appendix D: HIRO + Freeze**. In this variant, we first pre-train the low-level system and then freeze it while continuing high-level learning. This control gives HIRO an advantage because it provides the baseline with additional low-level pre-training. The results show that freezing alone does not explain our gain: on AntMaze, HIRO + Freeze yields only a small early benefit without a clear final improvement, and on AntFall it shows no meaningful advantage over standard HIRO. We therefore revised the manuscript to make our claim more precise. Our point is not that all non-stationarity disappears during training, but that, unlike conventional HRL, the planner is **not trained via RL on worker-induced high-level transitions**. Instead, it is optimized against a critic-defined geometric objective. Although the critic is still learned during training, low-level updates affect the planner only indirectly through the replay data used to improve the critic, rather than by directly changing the planner’s transition dynamics. We now state this distinction explicitly.
>
> **(2) The claim that “This principle is mathematically captured by the triangle inequality” needs proof or citation.**
>
> We agree that the original wording was too strong without attribution. We revised **Section 4.1** to explicitly cite **Liu et al. (2023)** for the triangle-inequality-type relation of the optimal goal-conditioned value function in sparse-reward goal-conditioned RL. We also clarify that our paper does not re-prove that result; rather, we use it as the geometric motivation for planning via intermediate subgoals. We further clarified the role of the learned critic. In our method, the MRN parameterization makes the negative critic output a quasipseudometric in the learned latent space by construction. Thus, the geometric structure used by the planner is architectural, not something assumed only after convergence.
>
> **(3) The paper should better motivate why a subgoal-based approach is needed, rather than directly learning a policy to the final goal.**
>
> We agree and revised the Introduction accordingly. We now explain explicitly that a direct goal-conditioned policy to the final goal must simultaneously solve **global routing** and **local control**. In the long-horizon and resource-constrained settings considered here, this can place a substantial burden on a single low-level controller. Intermediate subgoals reduce the effective control horizon of the worker while allowing the planner to capture global path structure.
>
> **(4) We also addressed the remaining presentation and notation issues raised by the reviewer.**
>
> We now reference Figure 1 in the Introduction, improved the visual quality of Figures 1 and 2, and made the notation in Section 3.1 more rigorous by explicitly defining the transition distribution and scalar-valued reward. We also clarified Section 3.4 by defining MLP at first use, making clear which parts summarize prior work, and adding the appropriate citation and motivation.
>
> Overall, we believe the revision substantially strengthens the paper by clarifying the scope of the non-stationarity claim, grounding the geometric argument more rigorously, improving the motivation for subgoal-based planning, including an additional HIRO + Freeze experiment in **Appendix D**, and tightening the exposition throughout.

---

> > ### Comment · Reviewer_VqMh · 2026-03-12
> >
> > I thank the authors for their response as well as the revised draft of the paper.
> >
> > In my original review, I had two major concerns:
> > - That the claim that “This principle is mathematically captured by the triangle inequality” needs to be proved by the authors or supported by a citation.
> > - To properly gauge the effectiveness of the proposed method as it relates to the non-stationary claims, the authors need to compare their method to a HRL baseline that also freezes the low-level system.
> >
> > I believe that the first major concern was addressed by the authors now that they cite Liu et al. (2023) and make the necessary clarifications in Section 4.1.
> >
> > With regards to the second major concern, I believe that the changes that the authors have made *partially* address this concern, but not entirely. In particular, although the new experiments in Appendix D show that freezing the low level system does not always yield a significant advantage, these results remain somewhat disconnected  from the proposed approach. That is, it would be more convincing if the authors included their proposed method in the experiments from Appendix D to show that 1) freezing the low-level system on its own does not yield a significant improvement (which the currently results already show), and 2) that freezing the low-level system in combination with their proposed method yields better performance. In other words, as it stands, we have one set of experiments (Appendix D) that show 1), and a different set of experiments (in the main body) that show 2). However, to fully address my concern, the authors need to show 1) + 2) in the same set of experiments.
> >
> > I again thank the authors for their response.

---

> > > ### Author Response · Authors · 2026-03-19
> > > **Author Response**
> > >
> > > We thank the reviewer for the helpful follow-up. We agree that a shared three-way comparison among our method, HIRO, and HIRO + Freeze provides a more direct test of whether freezing alone is sufficient to explain the gains. We therefore added such a comparison in the revision (**Appendix E**).
> > >
> > > We first clarify an important distinction. Our method is not a pre-train-and-freeze method. In HIRO + Freeze, the low-level policy is pre-trained and then fixed for the remainder of training. In our method, by contrast, the low-level policy and the critic continue to be updated throughout training and are frozen only temporarily during each high-level update. Training therefore remains joint through alternating updates, rather than relying on a permanently fixed low-level component.
> > >
> > > **Appendix D** was introduced to answer a narrower causal question: whether the gains of our method could be explained simply by the freeze-only intervention itself. To isolate that question, Appendix D uses a within-HIRO control that keeps the underlying HRL framework fixed and varies only the freezing intervention. **The results do not support the freeze-only explanation**: HIRO + Freeze does not recover effective learning in the resource-constrained setting, and it provides no consistent final benefit even with a capable worker.
> > >
> > > At the same time, we agree with the reviewer that a unified three-way comparison including our method, HIRO, and HIRO + Freeze is useful as a shared experimental view. We therefore added such a comparison in **Appendix E** under a shared sparse goal-conditioned reward setting. We chose this setting because it aligns with the regime studied in this paper and used to evaluate our HER-based method. Since the methods were not originally introduced under identical reward and training regimes, we present this unified comparison as supplementary rather than as a fully matched benchmark.
> > >
> > > The new unified comparison shows that, under this shared sparse setting, **neither HIRO nor HIRO + Freeze exhibits reliable learning**. In most environments, both remain near zero success throughout training. In PointMaze, HIRO + Freeze attains a small nonzero success rate from the beginning, but it shows no clear improvement over training, while HIRO fluctuates at a similarly low level. We interpret this as limited initial competence from pre-training rather than successful learning. By contrast, **our method learns reliably under the same sparse setting**.
> > >
> > > Taken together, **Appendix D and Appendix E** address the reviewer’s concern in two complementary ways. Appendix D isolates the freezing intervention within the HIRO framework, showing that **freezing alone is not sufficient**. Appendix E places all three methods in a shared training setting and shows that our method learns reliably, while HIRO and HIRO + Freeze do not. These results support the narrower claim we intend to make: the gains of our method cannot be attributed simply to freezing the low-level system, but to changing how the planner is optimized. Specifically, our method reduces planner-side non-stationarity by optimizing the planner against a critic-defined geometric objective, rather than via RL on worker-induced high-level transitions.
> > >
> > > We hope this additional comparison addresses the reviewer’s concern more directly.

---

> > > > ### Comment · Reviewer_VqMh · 2026-03-20
> > > >
> > > > I thank the authors for their most recent changes.
> > > >
> > > > As per Appendix E, my concern regarding the low-level policy freezing has been adequately addressed.
> > > >
> > > > I thank the authors for running the additional experiments which have significantly strengthened their claims.

---

### Review · Reviewer_GDfZ · 2026-02-11

**Summary Of Contributions:**

### Summary

In hierarchical RL, there are two types of policies; a planner policy that proposes subgoals and a worker that executes the goal-conditioned policy. During learning both the planner and the workers update their parameters which introduces non-stationarity.
The paper proposes to decouple the planner and the worker, and instead change the planner to depend on a critic network rather than the low-level workers. This proposal is motivated by a theoretical result in goal-conditioned RL, which states that the learned optimal action value has the properties of a distance measure. The paper also proposes a regularization loss for learning the critic. Finally, they evaluate their approach in a resource-constrained setting which is a suitable setting where a light-weight policy would be preferred.

### Key Strengths
- The paper is well written and easy to follow.
- There are applications where we might want to deploy a light-weight policy, and decoupling the planner from the worker can allow for that. Since the planner can be run on an external device with access to more compute, while the worker is deployed on a robot or other resource-constrained hardware.
- The experiments clearly demonstrate when we should expect (or not expect) an improvement from using the proposed method. As motivated in the introduction, the decoupling is better suited for low-resource constrained settings. It’s good though that the performance doesn’t degrade when the limited-resources condition is removed.

### Weaknesses/Questions:
- Until section 3.2, I was under the impression that the authors are using a fixed pre-trained action-value so that it provides a stable critic for the planner. However, in section 3.2 and 3.3, I see that Q is learned jointly with the planner. Now, i am wondering:
     - If Q is changing, why is basing the planner on this changing Q is better than basing it on the low-level worker? shouldn’t the fact that Q is changing introduce non-stationarity as well.
     - The triangle inequality is a property of the optimal action-value function, is it guaranteed to be true during learning the action-value?

- It seems that the regularization doesn’t improve over the hierarchy, do you have an explanation why that’s the case? The improvement in EggRotate and PointMaze are only marginal (the confidence intervals are overlapping so it’s hard to say it improves on the hierarchy)

**Audience:**

Yes

**Audience Explanation:**

Goal conditioned RL is of interest to the community of reinforcement learning researchers.

**Broader Impact Concerns:**

No ethical concerns.

**Claims And Evidence:**

Yes

**Claims Explanation:**

One of the motivations behind the decoupling of the planner and the worker was that we could only deploy a light-weight worker to settings with limited resources. The experiments in section 4.3.2 clearly demonstrate that the proposed approach works in such resource-constrained setting.

**Requested Changes:**

- Changes that answer/clarify the questions mentioned in Weaknesses/Questions section above.
- I think it's important to expand on the triangle inequality for the action-value function during learning. For example, is the Q function guaranteed to be a distance measure  during training? Are there non-stationarity introduced from jointly training the Q-function? If so, is that worse or better than when the planner was based on the low-level worker.

---

> ### Author Response · Authors · 2026-03-11
> **Author Response**
>
> We thank the reviewer for the careful reading and for raising these important points. We agree that our original presentation was not sufficiently clear about (i) the type of non-stationarity addressed by our method, (ii) the role of the triangle-inequality property during learning, and (iii) the intended effect of the regularization term. We revised the manuscript accordingly.
>
> **(1) If $Q$ changes during training, why is basing the planner on $Q$ better than basing it on the low-level worker?**
> We agree that our original wording could be misread as suggesting that the critic is fixed. This is not the case: the critic $Q_\\theta$ is learned jointly with the planner. However, the key point is that the resulting non-stationarity is qualitatively different from that in conventional HRL.
>
> In standard subgoal-based HRL, the high-level policy is trained via RL on worker-induced high-level transitions. As the low-level worker changes, the effective transition process seen by the planner also changes, so the planner is trained on an evolving MDP. In contrast, in our method the high-level policy is trained by maximizing the geometric objective in Eq. (1), rather than by RL on worker-induced high-level transitions. Although $Q_\\theta$ changes during training, low-level updates affect the planner only indirectly through the replay data used to improve the critic, rather than by directly changing the planner’s transition dynamics. We therefore do not claim that all non-stationarity disappears; rather, our method avoids **direct worker-induced transition non-stationarity at the planner level**. We revised the manuscript to make this distinction explicit (e.g., Section 4.1).
>
> **(2) The triangle inequality is a property of the optimal action-value function. Is it guaranteed during learning?**
> We appreciate this question and agree that the original text did not sufficiently distinguish the theoretical motivation from the structural property of the learned critic. We revised this point to make the distinction explicit.
>
> The triangle-inequality property of $-Q^\\star$ is the theoretical motivation. During learning, what is guaranteed is an **architectural property** of the MRN parameterization: for any parameters $\\theta$, the negative MRN output defines a quasipseudometric in the learned latent space. In other words, $-Q_\\theta$ satisfies the triangle inequality in the latent space by construction, even before convergence. This is not a claim that the critic has already converged to $Q^\\star$; rather, learning determines how well the learned geometry approximates the true optimal travel cost. We now state this more clearly in the manuscript (Section 3.4).
>
> **(3) The regularization seems to improve performance only marginally. Do the authors have an explanation?**
> We thank the reviewer for this important comment. We agree that the effect of the regularizer is task-dependent rather than uniformly large. In the revision, we changed the uncertainty reporting in the comparison plots from standard deviation to standard error, which is more appropriate for comparing mean performance across methods, and we softened the corresponding discussion in the text. Our view is that **the regularizer is particularly helpful in environments where preserving accurate global geometry is especially important for planning**, such as obstacle-rich tasks like PointMaze. By contrast, in environments where such global structure is less critical, we do not make a strong claim that the regularizer yields a large performance gain. We therefore revised the manuscript to present the effect of the regularizer more cautiously.

---

> > ### Comment · Reviewer_GDfZ · 2026-03-23
> >
> > Thank you to the authors for the thorough revision.
> >
> > The clarifications around the changing Q and the architecture property of MRN parametrization have addressed my concerns. I also find the revisions regarding these two points satisfactory.
> >
> > on regularization: I appreciate the softened language and the change to standard error. That said, the empirical results still doesn't show advantage for using the regularizer. I find it hard to reach the conclusion that the regularization is useful based on the current results.
> >
> > regarding other concerns raised during the discussion phase: the new related work section positions the paper relative to QRL and HIQL which was missing in the original submission.
> > I find the results in appendix F interesting, while it shows that if you use hierarchy you need MRN, it kind of undercuts the need for hierarchy over the baseline.
> >
> > I think the main advantage of the hierarchy shows only in the resource-constrained setting, and that is still a useful setting.
> >
> > Overall, I believe the revision has substantially addressed my concerns, and made the paper better than the original submission.

---

### Review · Reviewer_xy4S · 2026-02-26

**Summary Of Contributions:**

The paper presents an approach to hierarchical reinforcement learning that disentangles low level and high level policy. The high level policy is trained to output goal states via a Q function which is trained to obey the quasimetric structure of the trajectory space. The lower level policy then executes on close goals.

The authors validate their proposed architecture in several goal-reaching tasks, and provide ablations on the capacity of the low level worker.

**Audience:**

No

**Audience Explanation:**

The detailed answer is: Yes, this paper has an audience, but this audience will likely be familiar with methods like HIQL and quasimetric learning, and so those connections will have to be made before the paper will be interesting to this audience.

**Claims And Evidence:**

No

**Claims Explanation:**

The most critical issue with the paper is that it fails to engage with two vital pieces of the literature: Optimal Goal-Reaching Reinforcement Learning via Quasimetric Learning, Wang et al., https://www.tongzhouwang.info/quasimetric_rl/ and HIQL: Offline Goal-Conditioned RL with Latent States as Actions, Park et al., https://seohong.me/projects/hiql/ Both of these works are well established and recognized in the literature, and deal with important components of the presented framework. The Quasimetric learning paper tackles the metric learning part of the paper presented here, albeit without addressing hierarchical RL explicitly, and HIQL proposes a similar trajectory subgoal architecture as the presented paper here. Several more recent papers have recently started expanding on HIQL to integrate explicit planners.

However, I don't believe that the ideas presented in this paper are without merit or that the main claims are uninteresting. My negative rating lies solely on the fact that as it stands, by failing to compare and contrast to highly related works, it is hard to assess which of the proposed issues and solutions are still relevant problem.

The intuition on decoupling lower and higher level planning is strong, and I would actually encourage the authors to integrate Appendix B into the main paper directly after the Introduction and preliminaries. Having clean experiments that highlight the core problem in the main body of the paper is important to ground the claims the authors make.

Some minor nitpicks: The authors write "Simply forcing HIRO to adopt an MRN would constitute a major modification of the original framework and prevent a fair comparison". I believe this is a somewhat inadmissible claim, as the goal of research is not to show of different concrete algorithms, but to showcase which components of the algorithm are needed to solve which parts of the problem. This also leads over to the fact that I would love to see some more validation on how much the higher level planner benefits from the quasimetric structure. Given that HIQL proposes a similar setup, but without an explicit quasimetric loss, I think a very important set of experiments would be to assess if this explicit structure helps. Other variants that could be tried would be goal-contrastive approaches or even techniques like "Search on the replay buffer" if offline RL is an interesting target.

More detailed feedback:
- Overall the paper cites almost no work after 2020. While many papers err on the side of overly focusing only on recent published work, this paper does omit the massive body of literature established in goal-conditioned RL since 2020. I would strongly encourage the authors to conduct a thorough literature research. If the authors are interested, I am happy to give additional pointers to the ones above.
- An important ablation, as pointed out above, is to check if the metric space structure is necessary, of if the trajectory goal conditioning is sufficient on its own.
- Figure 5: Does the Hierarchy + Regularization also use 4 workers? Otherwise, I'm not sure what the graph is supposed to be validating. Given the text I assume yes, but the legend is confusingly worded.
- Nitpick: one standard deviation of the policy performance is not an interpretable measure of variation, unless the goal is to show the variability of each algorithm. For a comparison between algorithms in which this is likely supposed to indicate an uncertainty, it would be better to support something like 3 standard errors of the mean (if a normal assumption is valid) or a bootstrapped confidence interval. Given the relative large error bars in Figure 5, I am not sure all experiments have sufficient statistical power to support the claims.

**Requested Changes:**

I am unsure if these changes are within scope for a simple revision, but I will list them as encouragement to the authors to revise their paper. Overall, I like the established problem and framing, and I think this work can be very valuable.

- Review the recent literature on goal-conditioned RL, (quasi)metric learning, and hierarchical RL, with a special focus on the two papers mentioned above. Position this work clearly in this context.
- Provide additional experimental evidence about the impact of the metric space formulation of the high level planner, and compare to alternative approaches empirically or theoretically.
- Strengthen the exposition of the paper by clarifying the issue of entanglement and instability between low level and high level planner. Does the problem only arise with shared architectures, or are there other issues like shared data buffers.
- Provide cleanly interpretable visualizations of statistical uncertainty on measurements.
-

---

> ### Author Response · Authors · 2026-03-11
> **Author Response**
>
> We thank the reviewer for the careful reading and constructive feedback. We agree that the original submission did not sufficiently position the paper with respect to recent goal-conditioned RL and hierarchical RL literature, and that several core claims needed clearer support. We revised the manuscript accordingly.
>
> **(1) The paper should better position itself relative to recent literature, especially Quasimetric RL and HIQL.**
>
> We agree and added a new **Related Work** section (Section 2). In particular, we now discuss MRN, Quasimetric RL (QRL), HIQL, and related planning-based methods such as SoRB and HIGL, and clarify our position relative to them. Relative to MRN and QRL, our contribution is not a new quasimetric model or learning objective, but the use of an explicitly structured critic as a planning map within a hierarchical control architecture. Relative to HIQL, while both methods use intermediate subgoals, the difference is not only offline versus online: HIQL studies offline goal-conditioned RL via hierarchy extracted from a single learned goal-conditioned value function, whereas our paper studies online HRL and focuses on stabilizing high-level learning by avoiding direct RL updates on worker-induced high-level transitions.
>
> **(2) The paper should provide additional experimental evidence about the impact of the metric-space structure on hierarchical planning.**
>
> We agree and added a new ablation (**Appendix F**) that removes the explicit MRN metric structure while keeping the rest of the framework as unchanged as possible. Specifically, we replace the MRN-based critic with a critic defined on the concatenated latent pair, which can still serve as a learned value function but does not impose an explicit quasimetric or triangle-inequality structure. The results show a clear asymmetry: this critic can still support standard non-hierarchical goal-conditioned learning to some extent, but when used as the basis for our hierarchical planner, performance largely collapses and success remains near zero in most environments. This supports our main claim: in our framework, the critic is not merely a training signal but a planning substrate, and the explicit geometric structure is important for reliable subgoal planning.
>
> **(3) The paper should clarify what kind of entanglement / instability is being addressed.**
>
> We agree and strengthened the exposition of this point in the Introduction and Section 4.1. Our claim is not that the problem arises only from shared architectures, nor that all non-stationarity disappears in our method. The central issue in conventional HRL is that the planner is trained through **worker-induced high-level transitions**, so as the low-level policy changes, the effective high-level dynamics also change. This issue is not specific to shared architectures or shared data buffers. In contrast, our planner is optimized against a critic-defined geometric objective rather than through high-level RL on worker-dependent transitions. Thus, low-level updates affect the planner only **indirectly** through the critic-learning process, rather than by directly changing the planner’s transition dynamics.
>
> **(4) The paper should provide more interpretable uncertainty visualizations.**
>
> This is a helpful point. In the revision, we clarified the corresponding figure legends and captions, including the multi-worker setting in **Figure 5**, and changed the uncertainty reporting in the comparison plots from standard deviation to standard error, which is more appropriate for comparing mean performance across methods. We also adjusted the corresponding discussion to match the revised uncertainty reporting.
>
> Overall, we believe the revision substantially strengthens the paper by clearly positioning it relative to recent literature, adding direct evidence that the explicit metric structure is important for hierarchical planning, sharpening the claim about the specific form of non-stationarity addressed, and improving the interpretability of the empirical presentation.

---

> > ### Comment · Reviewer_xy4S · 2026-03-28
> > **Reviewer reply**
> >
> > Dear authors, please excuse my tardy reply. I have been travelling and dealing with some personal issues, and so I apologize for causing a delay for your paper.
> >
> > I have finally found the time to review the revised submission. I believe you have adequately addressed all of my major concerns, and I concur with the other reviewers that this has greatly strengthened the paper.
> >
> > Regarding the presentation of the paper, I would strongly advise the authors to place the low-resources experiments more prominently. Currently the more inconclusive results in Figure 3 are presented first, which leaves the first impression that your method does not matter much. It's likely safe to assume that is clearly not the lesson you want readers to take away from your work, so I would advise you to present these results with a stronger focus on the setting where your method does shine. It is always a good idea to present the strongest results first!

---

> > > ### Author Response · Authors · 2026-03-29
> > > **Author Response**
> > >
> > > We thank the reviewer for the thoughtful follow-up and for the positive assessment of the revision. We also appreciate the helpful suggestion regarding the presentation of the experimental results.
> > >
> > > We agree that the resource-constrained setting should be given greater prominence in the paper. In response, we reorganized the experiments section so that the resource-constrained results are presented before the capable-worker results. We also revised the corresponding text to clarify that the resource-constrained setting is the primary evaluation setting of the paper, while the capable-worker setting is included as an additional evaluation.
> > >
> > > We believe this reorganization improves the presentation and better aligns the structure of the paper with its central message. Thank you again for this valuable suggestion.

---

### Author Response · Authors · 2026-03-11
**Summary of Major Revisions**

We thank the reviewers for the constructive feedback. In response, we made the following major revisions:

* Added a **Related Work** section to clarify the paper’s relationship to recent goal-conditioned and hierarchical RL.
* Added **Appendix F**, a new ablation on the **explicit metric structure**, providing direct evidence that the geometric structure is important for hierarchical planning.
* Added **Appendices D and E**, introducing a new **HIRO + Freeze** control and a supplementary three-way comparison, which together provide stronger evidence that the observed gains are not attributable solely to freezing the low-level system.
* Clarified the paper’s claims about **non-stationarity** and **triangle inequality**, and improved the presentation of **uncertainty** and **figures**.

*Note*: Because we added a new Related Work section as Section 2 in the revised manuscript, some section numbers are shifted by +1 relative to those referenced in the original reviews.

---

### Decision · Action_Editor_HBEQ · 2026-04-16

**Recommendation:** Accept as is

**Audience:**

Yes

**Audience Explanation:**

With the revisions that better situate the work within the context of recent related work, the reviewers agree that the paper would be valuable to some of TMLR's readership, particularly those interested in hierarchical and/or resource-constrained RL.

**Claims And Evidence:**

Yes

**Claims Explanation:**

The primary claim of the paper is that, by reframing the task of the higher-level learner (the "planner") in hierarchical RL as navigation in a metric space, the learning problem becomes easier. The "planner" can learn a path that identifies subgoals for lower-level learners rather than learn a policy using the ever-changing lower-level policies as temporally extended actions. The reviewers noted similarity to prior work and questioned the conclusiveness of the results. After a robust discussion and a substantial revision, the reviewers agree that this claim is properly contextualized in the literature and well-supported by controlled experiments that isolate the contribution of the proposed method.

Secondary claims include the modest positive impact of a regularization method on the representation of the space and the benefit of this approach in low-resource settings (e.g. edge computing). The former is supported both by connection to existing theoretical results  in the literature and experimental results demonstrating that regularization results in higher overall performance in some tasks. The latter is supported by experiments demonstrating that, because low-resourced "worker" learners can have volatile behavior/performance, learning a higher-level "planner" policy directly using such policies as meta-actions can be difficult. The proposed method reduces this dependence, making the "planner" policy's learning task more stable and tractable.